

# Stationarization and Multithermalization in spin glasses

Pierluigi Contucci[1], Federico Corberi[2], Jorge Kurchan[3] and Emanuele Mingione[1]

**1** Università di Bologna, Italy
**2** Dipartimento di Fisica "E. R. Caianiello" and INFN, Gruppo Collegato di Salerno
and CNISM, Unità di Salerno, Università di Salerno,
via Giovanni Paolo II 132, 84084 Fisciano (SA), Italy
**3** Laboratoire de Physique de l'Ecole Normale Supérieure, ENS, Université PSL, CNRS,
Sorbonne Université, Université Paris-Diderot, Sorbonne Paris Cité, Paris, France

## Abstract

We develop further the study of a system in contact with a *multibath* having different temperatures at widely separated timescales. We consider those systems that do not thermalize in finite times when in contact with an ordinary bath but may do so in contact with a multibath. Thermodynamic integration is possible, thus allowing one to recover the stationary distribution on the basis of measurements performed in a 'multi-reversible' transformation. We show that following such a protocol the system is at each step described by a generalization of the Boltzmann-Gibbs distribution, that has been studied in the past. Guerra's bound interpolation scheme for spin-glasses is closely related to this: by translating it into a dynamical setting, we show how it may actually be implemented in practice. The phase diagram plane of temperature *vs* "number of replicas", long studied in spin- glasses, in our approach becomes simply that of the two temperatures the system is in contact with. We suggest that this representation may be used to directly compare phenomenological and mean-field inspired models. Finally, we show how an approximate out of equilibrium probability distribution may be inferred experimentally on the basis of measurements along an almost reversible transformation.



# 1 Introduction

Glasses are dynamical objects, the properties that are relevant for them such as large viscosity and aging, are essentially dynamical in nature. Somewhat surprisingly, a fruitful theoretical approach to them has been to study the proprieties of the energy landscape through the equilibrium properties: this led to the Parisi scheme, and the discovery of hierarchical organization of states-within-states that it implies. On the other side, the mean-field solution of the dynamics starting from a random configuration is largely well-understood: it is characterized by the emergence of widely separated timescales, each with a different characteristic temperature: a situation we shall denote as 'multi-thermalization'. The Parisi and dynamic multi-thermalization constructions are such that, even if we do not know the actual solution of any finite-dimensional glass model, we do know what both would imply for it.

The purpose of this paper is to establish a stronger connection between static solution and dynamic multi-thermalization. In section 2 we review the properties of a system in contact with a bath having different temperatures in the limit of widely-separated timescales: in short, a 'multibath'. We establish the notion of 'multi-thermalization', as the condition in which the system is stationary and has, for all its observables, the same fluctuation-dissipation temperature as the bath, this equality holding at each timescale. A system which would equilibrate with the fastest of these baths (a liquid, a paramagnet) will also multithermalize, but

the multibath will in itself generate some slow tails of correlation function syncronized with it. These clearly disappear when the coupling to the bath is weak. A more dramatic situation is known to arise in mean-field glass models: the system, starting form a high temperature configuration, never becomes stationary after being placed in contact with a low-temperaure bath: it 'ages'. Instead, a weak multibath may make it stationary: how weak it may be to do so depends on its timescale - the longer the timescale the weaker the bath needed. We discuss this situation in detail in Sections 2.1 and 2.2. Beyond mean field models, one may prove that the multi-thermalization situation is still valid, provided the system satisfies in equilibrium a Parisi scheme (although we do not have at present any model for which we may prove that this happens): we know this by extending trivially the result of Franz et al. [19]. This point will be discussed in Sec. 4.2. In Section 5 we show how to extend the classical thermodynamical notion of reversible transformation into a multi-reversible one. This allows us, both in theory and in practice, to transform multi-reversibly a system and then infer the probability distribution it follows at each step. We come back to this in Sec 8, where we show how this procedure may be implemented (at least numerically) in a simulation of a realistic structural glass. Thus, from the dynamic Fluctuation-Dissipation data one may reconstruct the distribution, which is a generalization of the Boltzmann-Gibbs one. Note that, for this to be the case, it is necessary that the system admits multi-thermalization at each step of the multi-reversible transformation. At this stage, one recognizes that the constructions we are using are closely related to the construction that Guerra [14] used to prove a bound on the free energy of the Sherrington-Kirkpatrick and other models. In Section 7, we uncover the dynamical content of Guerra's scheme: this gives a physically appealing – and numerically realizable – implementation of the procedure.

## 2 Multibath and thermalized disorder

In a spin-glass system such as the Sherrington-Kirkpartick(SK) model of spins $\sigma_i$ interacting through a random interaction $J_{ij}$

$$H(\sigma, J) = -\frac{1}{\sqrt{N}} \sum_{i,j=1}^{N} J_{ij}\sigma_i\sigma_j - \sum_{i=1}^{N} J_i\sigma_i, \tag{1}$$

one needs to compute the averaged logarithm of the partition function

$$\mathbb{E}\left\{\ln\left[\sum_{\sigma} e^{-\beta H(\sigma,J)}\right]\right\}, \tag{2}$$

where $\mathbb{E}$ denotes expectation with respect to both the two-body interactions $J_{ij}$, and the external magnetic fields $J_i$ . The former expression provides in fact the correct generating functional for the *quenched* moments of the Hamiltonian where the quenched measure is defined, for an observable $A$, as

$$\mathbb{E}\left[\frac{\sum_{\sigma} A(\sigma,J) e^{-\beta H(\sigma,J)}}{\sum_{\sigma} e^{-\beta H(\sigma,J)}}\right]. \tag{3}$$

In other words at fixed disorder $J$ a Boltzmann-Gibbs computation is made on $\sigma$ and later averaged on the disorder. The (easier to compute) *annealed* measure has instead a different physical significance, and corresponds on a standard Boltzmann-Gibbs computation on $(J, \sigma)$:

$$\frac{\mathbb{E}\left[\sum_{\sigma} A(\sigma,J) e^{-\beta H(\sigma,J)}\right]}{\mathbb{E}\left[\sum_{\sigma} e^{-\beta H(\sigma,J)}\right]}. \tag{4}$$

The need to calculate the difficult expression (2) gave rise to the replica computation

$$\frac{1}{n} \ln \mathbb{E}\left\{\left[\sum_\sigma e^{-\beta H(\sigma, J)}\right]^n\right\}, \qquad n = 1, 2, \dots, \tag{5}$$

inferring (and guessing with a suitable ansatz) the expression (2) by means of continuation for $n \to \zeta$, a real positive number, and its limit when $\zeta \to 0$. This was accomplished by Parisi in a remarkable series of papers [27, 32].

Integrals like (5) are ubiquitous in Parisi's construction, as intermediate steps, with generic $n$. One may ask if there is a more physical way to interpret them. Indeed this is so. As we shall see in detail below, if evolving the spins at temperature $T$, and, at a much slower rate, evolving the $J$'s at temperature $T/n$, we precisely obtain the averages generated by (5). This relation between a replica computation and a 'two-bath' computation is just one instance of a deeper and more general one [5, 6, 10, 13, 21, 36].

Consider a system with two sets of variables $\mathbf{x_1}$ and $\mathbf{x_2}$ and Hamiltonian $H(\mathbf{x_1}, \mathbf{x_2})$. We assume that the variable $\mathbf{x_2}$ reaches equilibrium with relaxation time $\tau_2$ while in contact with a thermal bath at temperature $T_2$. Similarly the variable $\mathbf{x_1}$ has relaxation time $\tau_1$ at temperature $T_1$.

In the limit $\tau_2 \ll \tau_1$ we may formalize an equilibrium theory for the system described by the following thermodynamic functions. Setting $\beta_1 = 1/KT_1$, $\beta_2 = 1/KT_2$ (where $K$ is the Boltzmann constant) the free energy is obtained in two steps :

$$F_1[\mathbf{x_1}] = -\frac{1}{\beta_2} \ln\left[\int d\mathbf{x_2}\, e^{-\beta_2 H(\mathbf{x_1}, \mathbf{x_2})}\right] \tag{6}$$

$$F_0 = -\frac{1}{\beta_1} \frac{1}{\zeta_1} \ln\left[\int d\mathbf{x_1} e^{-\beta_1 F_1(\mathbf{x_1})}\right] = -\frac{1}{\beta_2} \ln Z_0 \tag{7}$$

$$Z_0 = \left\{\int d\mathbf{x_1}\left[\int d\mathbf{x_2}\, e^{-\beta_2 H(\mathbf{x_1}, \mathbf{x_2})}\right]^{\zeta_1}\right\}^{1/\zeta_1} ; \quad \zeta_1 = \frac{\beta_1}{\beta_2}. \tag{8}$$

The previous thermodynamic expressions, leading to a *nested* Gibbs-Boltzmann structure, assume that one evolution is adiabatic with respect to the other and that both, the slow and the fast one, have enough time to reach equilibrium.

*Remarks:* For $T_1 = T_2$ this whole construction reduces to the standard Gibbs-Boltzmann measure since $\zeta_1 = 1$. Identifying $\mathbf{x_2}$ with $\sigma$ and $\mathbf{x_1}$ with $J$ and choosing $\zeta_1 = n$ we find (5). Moreover for real $\zeta_1$ we recover the quenched free energy in the limit $\zeta_1 = 0$ while $\zeta_1 = 1$ corresponds to the annealed case.

The construction may be generalized to an arbitrary number of timescales $r > 2$, with their corresponding variables and temperatures, such that each evolution is adiabatic with respect to the previous. Namely we consider an Hamiltonian $H(\mathbf{x_1}, \mathbf{x_2}, \dots, \mathbf{x_r})$ where the degree of freedom $\mathbf{x_a}$ has relaxation time $\tau_a$ and is contact with a bath $\beta_a$. Assuming widely separated timescales $\tau_r \ll \tau_{r-1} \dots \ll \tau_1$ one obtain the full measure recursively. In terms of free energy starting from $F_r = H$ we define

$$e^{-\beta_a F_{a-1}} = \int d\mathbf{x_a}\, e^{-\beta_a F_a}, \tag{9}$$

for any $1 \le a \le r$. Defining for any $a = 1, \dots, r$ the parameter $\zeta_a = \beta_a/\beta_r$ then the free energy at the final step can be written as

$$F_0 = -\frac{1}{\beta_r} \ln Z_0, \tag{10}$$

where

$$Z_0 = \left\{ \int d\mathbf{x}_1 \left[ \int d\mathbf{x}_2 \dots \left[ \int d\mathbf{x}_r \, e^{-\beta_r H(\mathbf{x}_1, \mathbf{x}_2, \mathbf{x}_3, \dots, \mathbf{x}_r)} \right]^{\zeta_{r-1}/\zeta_r} \dots \right]^{\zeta_1/\zeta_2} \right\}^{1/\zeta_1}. \tag{11}$$

It is straightforward to show that the equivalent iterative expression for the *generating functional* (also called *pressure* in the mathematical literature):

$$P_a = -\beta_r F_a =: \ln Z_a \tag{12}$$

turns out to satisfy

$$e^{\zeta_a P_{a-1}} = \int d\mathbf{x}_a \, e^{\zeta_a P_a}. \tag{13}$$

We will call *multibath measure* the measure induced by the generating functional $P_0$.

Generating functionals of this kind were introduced by Parisi and Virasoro [31], without connection to any dynamics, as a concrete way to construct order parameters conjugate to replica symmetry breaking. It was later discussed in a dynamic context in [9, 13], and more recently in [5]. Moreover the recursion (13) is the core of Guerra's interpolation scheme [14] (see section 7). We shall derive dynamically the above measure in detail in two examples below.

## 2.1 Thermalized external fields

Referring to the above notation we will consider in this section $\mathbf{x}_1, \dots, \mathbf{x}_r$ as magnetic fields and $\mathbf{x}_{r+1}$ as the spin variables. To this purpose we consider a system made of $N$ interacting spins $\sigma = (\sigma_i)_{i \leq N}, \sigma_i \in \mathbb{R}$, with quenched two-body coupling $(J_{ij})_{i,j \leq N}$ and also coupled with a family of dynamic external fields $(J_i^a)_{i \leq N}^{a \leq r}$ through the Hamiltonian

$$H(\sigma, J) = -\gamma \sum_{i,j} J_{ij} \sigma_i \sigma_j - \sum_{a,i} \gamma_a J_i^a \sigma_i + \frac{A}{2} \sum_i (\sigma_i^2 - 1)^2 + \frac{1}{2} \sum_{a,i} (J_i^a)^2, \tag{14}$$

where $\gamma, \gamma_a$ are non negative real parameters, $A$ is a large positive constant that forces $\sigma_i \sim \pm 1$.

We use the notation $J = (J^1, \dots, J^r)$ and $J^a = (J_i^a)_{i \leq N}$ for any $a = 1, \dots, r$. In the Hamiltonian we kept the explicit dependence of only the dynamical variables $(\sigma, J)$ for fixed realization of $(J_{ij})_{i,j \leq N}$.

The main assumption on the dynamics is that the degrees of freedom $(\sigma, J)$ have widely separated timescales: denoting by $\tau_\sigma, \tau_a$ the relaxation times of $\sigma$ and $J^a$ respectively, we assume that $\tau_1 >> \tau_2 >> \dots >> \tau_r >> \tau_\sigma$. Clearly these scales may depend on the size of the system and become infinite as $N \to \infty$. The dynamic is described by a system of $r + 1$ Langevin equations:

$$\tau_\sigma \dot{\sigma}_i = -\frac{\partial H}{\partial \sigma_i} + \eta_i \quad \text{with} \quad \langle \eta_i(\tau) \eta_j(\tau') \rangle = 2T \tau_\sigma \delta_{ij} \delta(\tau - \tau') \tag{15a}$$

$$\tau_a \dot{J}_i^a = -\frac{\partial H}{\partial J_i^a} + \rho_i^a \quad \text{with} \quad \langle \rho_i^a(\tau) \rho_j^b(\tau') \rangle = 2T_a \tau_a \delta_{ij} \delta_{ab} \delta(\tau - \tau'), \ a \leq r. \tag{15b}$$

From now on we will set the Boltzmann constant equal to 1. Here $T = \frac{1}{\beta}$ and $T_a = \frac{1}{\beta_a}$, for $a = 1, \dots, r$, are the temperatures of the thermal bath of the spins and the fields respectively. For all practical purposes one can also think $\sigma_i$ evolving instead following a Glauber or Monte

Carlo dynamic with energy (14).

Our aim is to show that the stationary measure of such as dynamical system coincides with the multibath previously introduced. Let us start with the case $r = 1$. The spins have temperature $T_2 = T$ and timescale $\tau_\sigma$, in addition there is a single family of external fields $J \equiv J^1$ at temperature $T_1 = T'$ with timescales $\tau_1 \gg \tau_\sigma$. On a short time-scale compared to $\tau_1$ the spins evolve while the $J$'s are nearly constant. Hence on timescales $\tau_1 \gg \tau \gg \tau_\sigma$ the solution of (15a) is the usual Gibbs measure given $J$:

$$\mu(\sigma|J) = \frac{e^{-\beta H(\sigma, J)}}{Z(J)}. \tag{16}$$

On the other hand, (15b) is linear, and its solution is

$$J_i(\tau) = \int_{-\infty}^{\tau} \frac{d\tau'}{\tau_1} e^{-\left(\frac{\tau - \tau'}{\tau_1}\right)} \left[\gamma_1 \sigma_i(\tau') + \rho_i(\tau')\right]. \tag{17}$$

Using the assumption of adiabaticity, we may substitute $\sigma_i$ by its fast-time average given by the measure (16):

$$\sigma_i \longrightarrow \int d\sigma \, \mu(\sigma|J)\sigma_i = \frac{1}{\gamma_1}\left(T\frac{\partial}{\partial J_i} \ln Z(J) + J_i\right), \tag{18}$$

which depends on time through $J_i$. We get:

$$J_i(\tau) = \int_{-\infty}^{\tau} \frac{d\tau'}{\tau_1} e^{-\left(\frac{\tau - \tau'}{\tau_1}\right)} \left[T\frac{\partial}{\partial J_i} \ln Z(J) + J_i + \rho_i(\tau')\right]. \tag{19}$$

We now use the identity

$$\left(\tau_1 \frac{\partial}{\partial \tau} + 1\right)\left[\frac{1}{\tau_1} e^{-\left(\frac{\tau - \tau'}{\tau_1}\right)}\theta(\tau - \tau')\right] = \delta(\tau - \tau') \tag{20}$$

to transform equation (19) into:

$$\left(\tau_1 \frac{\partial}{\partial \tau} + 1\right)J_i = \left[T\frac{\partial}{\partial J_i} \ln Z(J) + J_i + \rho_i(\tau)\right]. \tag{21}$$

This is a Langevin equation with temperature $T'$ and potential $-T \ln Z$. In equilibrium, it leads to the distribution:

$$\mu(J) = \mathcal{N}e^{\beta' T \ln Z(J)} = \mathcal{N}[Z(J)]^\zeta, \tag{22}$$

where $\mathcal{N}$ is the normalization factor and $\zeta = T/T'$. Thus one obtains the *multibath measure* generated by (8)

$$\mu(\sigma, J) = \mu(J)\mu(\sigma|J) = \frac{Z(J)^\zeta}{\int dJ Z(J)^\zeta} \frac{e^{-\beta H(\sigma, J)}}{Z(J)}. \tag{23}$$

Notice that the term $\frac{1}{2}(J_i^1)^2$ in (14) carries in (23) as a centered Gaussian measure with variance $T'$. The general case with several $J_i^a$ with nested timescales is obtained by iteration, i.e. keeping at each step some variables as constants, and identifying the conditional distribution $\mu(J^a|J^{a-1}, ..., J^1)$. For a given $a$ the free energy $F_a = -\frac{1}{\beta} \ln Z_a$ acts as a potential for $J_i^a$, in the sense that in (15b) one can make the substitution

$$-\frac{\partial H}{\partial J_i^a} \longrightarrow \frac{1}{\beta} \frac{\partial}{\partial J_i^a} \ln Z_a. \tag{24}$$

Therefore one obtains

$$J_i^a = \int_{-\infty}^{\tau} \frac{d\tau'}{\tau_a} e^{-\left(\frac{\tau-\tau'}{\tau_a}\right)} \left[\frac{1}{\beta}\frac{\partial}{\partial J_i^a} \log Z_a + J_i^a + \rho_i^a(\tau')\right]$$

$$\left(\tau_a\frac{\partial}{\partial\tau} + 1\right) J_i^a = \left[\frac{1}{\beta}\frac{\partial}{\partial J_i^a} \ln Z_a + J_i^a + \rho_i^a(\tau')\right] \quad \text{given} \quad J^{a-1}, \dots, J^1 \quad \text{constant} \quad (25)$$

again a Langevin equation, which leads to the conditioned equilibrium $\mu(J^a | J^{a-1}, ..., J^1)$.

## 2.2 Correlation and Response for the multibath.

Another way to express the dynamics is to transform the problem with extra fields into a problem with no fields, in contact with a multibath. We start by writing (15b)

$$J_i^a = \int_{-\infty}^{\tau} \frac{d\tau'}{\tau_a} e^{-\left(\frac{\tau-\tau'}{\tau_a}\right)} \left[\gamma_a\sigma_i(\tau') + \rho_i^a(\tau')\right] = \gamma_a \int_{-\infty}^{\tau} d\tau' \, \mathcal{R}_a(\tau-\tau')\sigma_i(\tau') + \hat{\rho}_i^a, \quad (26)$$

where we have defined

$$\mathcal{R}_a(z) = \frac{1}{\tau_a} e^{-\frac{z}{\tau_a}} \theta(z), \quad (27)$$

$z = \tau - \tau'$, and

$$\hat{\rho}_i^a = \int_{-\infty}^{\tau} \frac{d\tau'}{\tau_a} e^{-\left(\frac{\tau-\tau'}{\tau_a}\right)} \rho_i^a(\tau') \quad \text{with} \quad \langle\hat{\rho}_j^b(\tau)\hat{\rho}_i^a(\tau')\rangle = \delta_{ij}\delta_{ab}\,\mathcal{C}_a(\tau-\tau'), \quad (28)$$

where $\mathcal{C}_a(z) = T_a e^{-\frac{|z|}{\tau_a}}$. Response and correlation satisfy the identities

$$\left(\tau_a\frac{\partial}{\partial z} + 1\right)\mathcal{R}_a(z) = \delta(z) \quad ; \quad \left(-\tau_a^2\frac{\partial^2}{\partial z^2} + 1\right)\mathcal{C}_a(z) = 2\delta(z)\tau_a T_a. \quad (29)$$

We notice that multibath acts on the spin dynamics as a memory term [10]. Indeed for (15a) one has:

$$\tau_\sigma\dot{\sigma}_i = \gamma\sum_j J_{ij}\sigma_j - A(\sigma_i^2 - 1) + \eta_i + \int_{-\infty}^{\tau} d\tau' \, \mathcal{R}(\tau-\tau')\sigma_i(\tau') + \hat{\rho}_i(\tau), \quad (30)$$

where $\mathcal{R}(z) = \sum_a \gamma_a^2 \mathcal{R}_a(z)$, the memory kernel, is the response function of the bath. The combined noise $\hat{\rho}_i(\tau) = \sum_a \gamma_a\hat{\rho}_i^a(\tau)$ is correlated as

$$\langle\hat{\rho}_i(\tau)\hat{\rho}_j(\tau')\rangle = \delta_{ij}\mathcal{C}(\tau-\tau') \quad \text{with} \quad \mathcal{C}(z) = \sum_a \gamma_a^2\mathcal{C}_a(z) \equiv \sum_a \gamma_a^2 T_a e^{-\frac{|z|}{\tau_a}}, \quad (31)$$

which defines the correlation of the multibath. Inserting (27) in the definition of $\mathcal{R}$ and comparing with (31), we obtain for the correlation and response of the multibath the relation

$$T(z)\mathcal{R}(z) = -\frac{\partial\mathcal{C}}{\partial z}\theta(z), \quad (32)$$

where $T(z)$ is constant within every one of the nested scales: $T(z) = T_a$ for $\frac{z}{\tau_a}$ away from zero and of order one to stay within the time-scale. The function $T(z)$ defines the *effective temperature* of the multibath [11–13, 21] Since $\beta_a = \zeta_a\beta$ then one can write the inverse effective temperature $\beta(z) = 1/T(z)$ as

$$\beta(z) = \beta x(z) \quad \text{with} \quad x(z) = \zeta_a \text{ if } \frac{z}{\tau_a} \sim 1. \quad (33)$$

Notice that the function $x$ can be viewed as the dynamical analogous of the Parisi order parameter in Guerra's bound as explained in detail in section 7. The physical reason for the fact that effective temperatures increases with increasing timescales was discussed by [6].

Here it is important to remark that although we used a particular set of couplings, leading to exponential decays in time, any bath with this fluctuation-dissipation relation and nested timescales will do for the purposes of this paper.

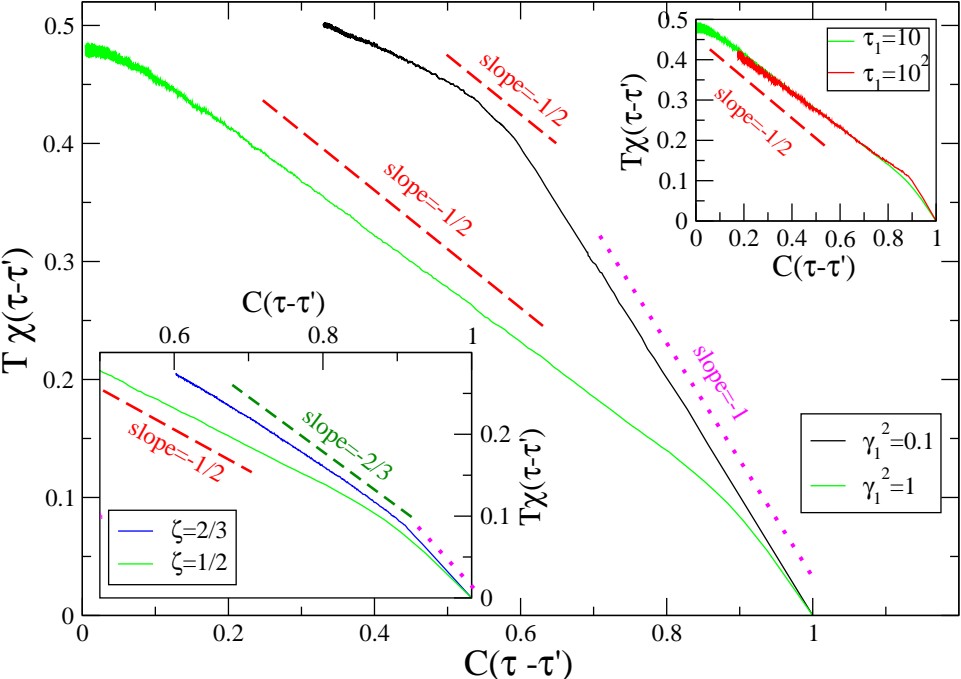

Figure 1: The fluctuation-dissipation plot of the model with multithermalized external field $J_i$. The integrated response $T\chi(\tau,\tau')$ is plotted against the autocorrelation function $C(\tau,\tau')$ for a system of $N = 2^{10}$ spins with random bimodal coupling constants $J_{ij} = \pm 1/\sqrt{N}$. The external fields are bimodal variables $\pm\sqrt{\gamma_1^2}$. The temperature of the spins is $T = 1/2$ and that of the magnetic field is is $T' = T/\zeta$, with $\zeta = 1/2$. Curves averaged over $8 - 19 \cdot 10^3$ realisations, for the various cases (in the main figure and in the insets). The perturbing field for the computation of $\chi$ is set to $H = 0.1$. The timescale of the slow bath is $\tau_1 = 10$ for all the curves. $\tau'$ is set to $\tau' = 1500$ in order to be in the stationary state. The dotted line is the equilibrium slope $-1$ while the dashed red one are the expected slope $-1/2$. Lower inset: comparison, for $H = 1$, between the case with $\zeta = 1/2$ (green curve, same as in the main figure) and $\zeta = 2/3$ (blue curve). For the latter it is $\tau' = 600$. Upper inset: comparison, for $H = 1$ and $\zeta = 1/2$, between the case with $\tau_1 = 10$ (green curve, same as in the main figure) and with $\tau_1 = 100$. For the latter it is $\tau' = 6000$.

## 2.3 The SK model with a multithermalized random field

In this section we briefly discuss the equilibrium properties of the static analogous of the Hamiltonian (14), i.e. a SK model with gaussian external fields coupled with different thermal baths. The Ising spins case has already been introduced in [31] where a Parisi-like formula has been obtained within the replica framework. More recently the solution has been extended and rigorously proved for generic spin distribution showing an interesting link with Hamilton-Jacobi PDE framework [23]. Here, we briefly review the simpler case of Ising spins and gaussian external field coupled with an additional thermal bath ($r=1$). The Hamiltonian of the system can be written as

$$H_N(\sigma) = -\frac{\gamma}{\sqrt{N}} \sum_{i,j} J_{ij}\sigma_i\sigma_j - \sum_{a,i} \gamma_a J_i^a \sigma_i, \tag{34}$$

where $\sigma_i = \pm 1$ and the $J$'s are independent standard Gaussian random variables. The Hamiltonian (34) can be seen as the static analogous of (14) where we take Ising spins and the Gaussian weight is absorbed in the distribution of the $J$'s variables. The factor $\frac{1}{\sqrt{N}}$ in (34) ensures a well defined thermodynamic limit. Notice that $H_N(\sigma)$ is a Gaussian process with covariance

$$\mathbb{E} H_N(\sigma^1)H_N(\sigma^2) = N \left[ \gamma^2 (q_{12})^2 + \gamma_1^2 q_{12} \right], \tag{35}$$

where $q_{12} = \frac{1}{N}\sum_i \sigma_i^1\sigma_i^2$ is the overlap. The generating functional of the multibath measure (23) (also called *pressure density*) is

$$p_N = \frac{1}{\zeta N} \mathbb{E}_0 \log \mathbb{E}_1 (Z_N)^\zeta, \tag{36}$$

where $\mathbb{E}_{0,1}$ denotes the average w.r.t the quenched coupling and the fields respectively and

$$Z_N = \sum_\sigma e^{-\beta H_N(\sigma)}. \tag{37}$$

If $\zeta_0 \to 0$ we recover the SK model in a quenched random field with variance $\gamma_1^2$. For real $0 < \zeta < 1$ one can prove that, following the same procedure presented in [4], the limiting value may be represented as a Parisi-like variational problem:

$$\lim_{N\to\infty} p_N = \inf_{x\in X_\zeta} \mathcal{P}(x), \tag{38}$$

where

$$\mathcal{P}(x) = \log 2 + f(0,h;x) - \frac{\beta^2\gamma^2}{2} \int_0^1 x(q)q\,dq, \tag{39}$$

and $f(q,y;x)$ satisfy a suitable Parisi's PDE. The infimum is taken over $x \in X_\zeta$ which is the space of distribution functions containing the point $\zeta$ in the image. This constraint is due to the fact that external field is not quenched and it is averaged out according to (36). The optimal $x$ solution of (38) represents the limiting distribution of the overlap w.r.t. the multibath measure induced by (36).

In order to explain the relation between the multibath measure and the dynamics described in the previous section we recall the definition of effective temperature. Given the two-time correlations $C(\tau)$ of the spins and the associated response $R(\tau)$ and integrated response $\chi(\tau) = \int_0^\tau R(\tau-\tau')d\tau'$ then the *effective temperature* can be defined by the relation [8,12,17]

$$\frac{1}{T(z)} = \beta(z) = -\frac{d\chi(z)}{dC(z)}. \tag{40}$$

Now we consider the integrated response as function of the correlation in the large $N, \tau$ limit

$$\lim_{\substack{z \to \infty \\ C(z)=C}} \chi(z) = \chi(C). \tag{41}$$

It has been proved in [19] that for a class of spin glass models *in finite dimensions* and under suitable assumptions (see the discussion in section 4.2) the quantity $\chi(C)$ provides a direct link between static and dynamics, more precisely

$$\beta \bar{x}(q)\big|_{q=C} = -\frac{d\chi(C)}{dC}, \tag{42}$$

where $\bar{x}$ is the distribution of the overlap of the system w.r.t. the equilibrium measure.

We have studied this dynamical quantities by means of numerical simulations, which we now detail (such description applies to the simulations of Secs. 2.4,2.5,3.1 as well). We considered a system with bimodal couplings $J_{ij} = \pm 1/\sqrt{N}$ and similarly for the fields $J_i = \pm\sqrt{\gamma_1^2}$. Such bimodal forms are chosen for numerical convenience, we don't expect major differences with respect to the Gaussian case considered insofar. We set $N = 10^{10}$ (except for the data in Fig. 4 where $N = 10^{12}$). We have checked that larger values of $N$ do not yield substantial differences. Starting with a random initial distribution of the spins and of the $J$'s the system is evolved in time with Montecarlo rules using Glauber transition rates: a spin $\sigma_i$ is chosen at random and flipped with probability $w(\sigma_i \to -\sigma_i) = (1/2)(1 - \tanh \Delta E / T)$. The same transition rate is used for updating the fields $J_i$, with a different temperature $T' = T/\zeta$ and an extra factor $\tau_1^{-1}$ realising the slower evolution. The dynamical average $\langle \ldots \rangle$ is taken over a large number of initial configurations and thermal histories, namely Montecarlo trajectories. Due to this large averaging procedure statistical errors on the data are rather small, typically of the order of the symbols used to draw the data. The most significative source of errors is represented by systematic effects due, e.g., to finite times, specifically $\tau_1$. The response function is computed routinely by running in parallel a copy of the original system perturbed by a small (in principle $H \to 0$) external magnetic field $H$ (constant is space and time). Since data get noisier for smaller values of $H$ we fix the perturbation to the largest value above which we start seeing a dependence of the results on $H$. Stationarization is achieved by waiting a sufficiently long time such that both $C$ and $\chi$ are observed to depend only on the time difference $\tau - \tau'$. Specific values of the various parameters are given in the captions of the figures.

The result for $T\chi$ *vs* $C$ are shown in Fig. 1. In main part of the figure a comparison is shown between two cases with a relatively strong and weak field intensity, $\gamma_1 = 1$ and $\gamma_1 = 0.1$, respectively, for a given value ratio $\zeta = 1/2$ between the spin and field temperatures. In the case with strong field intensity $\gamma_1^2 = 1$, after a short FDT regime for $C \simeq 1$ a constant slope roughly of order $-\zeta$ is approximately observed in a wide range of $C$. For smaller values of $\gamma_1$ such slope is present only in a narrow range of $C$ after the FDT regime. In the lower inset we compare two cases with strong field but with two different values of $\zeta$. We observe that in both cases the slope of the approximately linear part to the left of the FDT regime roughly agrees with the values of $\zeta$. Fitting in the range $x \in [0.6, 0.8]$ we find slopes $-0.6$ and $-0.5$ for $\zeta = 2/3$ and $1/2$, respectively. In the upper inset we compare two cases with strong field and the same value of the temperatures ($\zeta = 1/2$) but with two different timescales $\tau_1$ of the slow bath. The two curves behave similarly, but the transition from the FDT slope to the non-trivial one on the left part of the plot is more sharp for larger $\tau_1$.

Assuming that the static-dynamic correspondence discussed above holds also for SK, these features can be related to the overlap distribution properties. For $\gamma_1 \ll \gamma$, i.e. weak external field, we expect that the system and its overlap distribution behaves like a standard SK model at inverse temperature $\beta\gamma$. Similarly for $\gamma \ll \gamma_1$, or equivalently weak two-body coupling, the system is driven mostly by the multibath measure induced by the external field. Here we

expect that the overlap distributions develops a plateau at height $\zeta$, or equivalently $T\chi$ vs $C$ has slope $-\zeta$ at the origin. Besides that, Figure 1 shows that, for finite $\gamma_1$ the extension of the plateau is an increasing function of $\gamma_1$.

## 2.4 Thermalized interaction parameters

It is also possible that the set of interaction parameters themselves are slow, multithermalized variables [5, 15, 26]. Referring to our general notation we choose $\mathbf{x}_1 = (J_{ij})_{i,j \leq N}$ and $\mathbf{x}_2 = (\sigma_i)_{i \leq N}$ and the Hamiltonian function

$$H(\sigma, J) = -\gamma \sum_i J_{ij}\sigma_i\sigma_j + \frac{k}{2}\sum_{ij} J_{ij}^2, \tag{43}$$

for some $\gamma, k > 0$. The interacting part of (43) is the same of an SK model but here the $J_{ij}$ are not quenched variables but evolve in time as the spin variables . The $\sigma_i$ follows a Langevin dynamic at temperature $T$ and energy (43) and the $J_{ij}$ evolve with a slower timescale at a different temperature $T'$:

$$\tau_1 \dot{J}_{ij} = -k J_{ij} + \gamma \sigma_i \sigma_j + \rho_{ij}(\tau). \tag{44}$$

The noise $\rho_{ij}$ is centered with variance $2T'$ , where $T'$ is the temperature of the second equilibrium bath. We write, as before:

$$J_{ij}(\tau) = \int^{\tau} \frac{d\tau'}{\tau_1} e^{-\frac{k}{\tau_1}(\tau-\tau')}\left[\gamma\sigma_i(\tau')\sigma_j(\tau') + \rho_{ij}(\tau')\right]. \tag{45}$$

If $\tau_1 \gg \tau \gg \tau_\sigma$ we may replace $\sigma_i(\tau')\sigma_j(\tau')$ with its average respect to the stationary $\mu(\sigma|J)$ defined as in (16):

$$\gamma\sigma_i(\tau')\sigma_j(\tau') \to \gamma\langle\sigma_i\sigma_j\rangle_J = \frac{1}{\beta}\frac{\partial}{\partial J_{ij}}\log Z(J) + k J_{ij}. \tag{46}$$

Therefore the solution of (45) gives the stationary measure for the $J$:

$$\mu(J) = \mathcal{N}[Z(J)]^\zeta, \tag{47}$$

where $\mathcal{N}$ is the normalization, matching the definition of multibath measure (23) with $T/T' = \beta'/\beta = \zeta$. The generating functional of the measure is

$$p_N = \frac{1}{\zeta N}\ln\int dJ\,[Z(J)]^\zeta = \frac{1}{\zeta N}\ln\mathbb{E}_0\left(\int d\sigma\, e^{\beta\gamma\sum_{ij}J_{ij}\sigma_i\sigma_j}\right)^\zeta, \tag{48}$$

where $\mathbb{E}_0$ denotes the average w.r.t. $J_{ij}$ that are independent Gaussian random variables with variance $T'/k$. Taking $\gamma = \frac{1}{\sqrt{N}}$ and $k = T'$ one gets the standard setting of the SK model. If $\zeta \to 0$ then (48) gives the quenched pressure in the same spirit of the replica trick (5). The thermodynamic limit of (48) for $\zeta > 0$ has been rigorsly studied in [4, 26]. It turns out that (48) is represented as a Parisi-like problem of the form (38) where the infimum is taken over the space of distribution functions that have a jump discontinuity with gap $\zeta$ at the origin. Notice that this constraint is harder then than the one obtained in the multithermalized external field case. The reason is that here there are no quenched variables coupled with the spins. Keeping in mind the connection between statics and dynamics provided by (40) and (42), we can conclude that the integrated response function reaches the origin with slope $\zeta$ . *In simple words, the fact that interactions are at temperature $T'$ has the effect of 'killing' all effective temperatures $T_{eff} > T'$ in the original problem.* This means that in the fluctuation-dissipation

diagram $T\chi$ vs $C$ the branch to the left of the point with slope $\zeta$ is straight with slope $\zeta$. We have checked this fact with numerical simulations, which have been previously detailed in Sec. 2.3. The only difference is that what is coupled to the slow bath are the coupling constants $J_{ij}$, and the fields $J_i$ are set to zero. The result of the simulations are showed in Figure 2. In the main figure, curves for $\tau_1 = 10^2$ and three choices of $\zeta$ are shown. We have checked that different values of the spin temperature $T$ and of $\zeta$ yield similar results. The overall behavior is similar to the one observed in Fig. 1, with an FDT part of slope 1 on the right sector of the plot and a different slope on the left. However in this case one observes much nicer straight behaviors in the latter sector and, in addition, the agreement between the observed slope and the expected one (i.e. $-\zeta$) is much better, except for $\zeta = 1/4$. However, for this value of $\zeta$, the comparison shown in the inset between the cases $\tau_1 = 10^2$ and $\tau_1 = 10^3$ indicates that the discrepancy is due to an insufficient value of $\tau_1$ and that there is a convergence to the expected slope increasing $\tau_1$. Fitting the data (using $\tau_1 = 10^3$ for $\zeta = 1/4$) in the range $x \in [0, 0.1]$ we find slopes $-0.68$, $-0.54$, and $-0.38$, for $\zeta = 2/3, 1/2$, and $1/4$, respectively. The curves are straight lines with good approximation: indeed the fitted slope is rather stable upon changing the fit interval in the range $x \in [0, 0.4]$, the difference being on the third significant figure.

## 2.5 Timescales of a system

Let us take advantage of this construction to discuss the question of timescale within a system. Let us consider the behavior of the autocorrelation function $C$ which, for the SK model thermalized couplings $J_{ij}$, is shown in Fig. 3 in the case with $T' = \infty$. Suppose the decay up to a plateau $q_{EA}$ is independent of $\tau_1$ (for large enough $\tau_1$): $q_{EA}$ is then defined as the Edwards-Anderson parameter for the multithermalized system. For correlations below this value, the decay scales non trivially with $\tau_1$. One possibility is:

$$C(\tau - \tau') \sim h\left(\frac{\tau - \tau'}{a(\tau_1)}\right), \quad \text{for} \quad C < q_{EA}, \tag{49}$$

for $a(\tau_1)$ a suitable growing function of $\tau_1$. We have put to the test the scaling of $C$ by means of numerical simulations. The results for the SK model with spin temperature $T = 1/2$ and coupling constants $J_{ij}$ coupled to a slow bath at temperature $T' = \infty$ are reported in Fig. 3. In the upper left panel we see that, for the SK model, the scaling (49) does not work at all. A scaling that indeed seems to work is the following:

$$C(\tau - \tau') \sim g\left(\frac{\ln(\tau - \tau')}{b(\tau_1)}\right), \quad \text{for} \quad C < q_{EA}, \tag{50}$$

for $b(\tau_1)$ another suitable growing function of $\tau_1$, and $g$ is a *decreasing* function [22]. This can be observed in the upper right panel of Fig. 3. It turns out that $b(\tau_1)$ increases as the logarithm of $\tau_1$, see inset and fit described in the caption.

To understand the meaning of this, following [8] we consider triangles of correlations at three large times $t_1 < t_2 < t_3$. Scaling (49) implies:

$$C(t_3 - t_1) = h\left\{h^{-1}[C(t_3 - t_2)] + h^{-1}[C(t_2 - t_1)]\right\} \tag{51}$$

an isomorphism of the sum. The range of values where the 'triangle relation' takes this form is usually called "a timescale', because all times involved are commensurate.

Instead, scaling (50) is:

$$C(t_3 - t_1) = g\left[\frac{1}{b(\tau_1)}\ln\left(e^{b(\tau_1)g^{-1}[C(t_3 - t_2)]} + e^{b(\tau_1)g^{-1}[C(t_2 - t_1)]}\right)\right]$$
$$\rightarrow_{\tau_1 \to \infty} \min\{C(t_3 - t_2); C(t_2 - t_1)\}, \tag{52}$$

which is ultrametricity in time: there are infinitely many timescales. In reference [8] a complete classification of all possibilities for the triangle relations of the form $C(t_3, t_1) = f(C(t_3, t_2), C(t_2, t_1))$ is made, based on the fact that in general $f$ must be an associative function.

Notice that the linear scale of the $y$ axis in the two upper panels of Fig. 3 is only suited for the inspection of the scaling properties for relatively large values of $C$. For the smallest values of correlation, using a logarithmic $y$ scale ( lower panel) allows one to appreciate that, for $C(\tau - \tau') \ll q_{EA}(\tau_1)$, there is a scaling form $C(\tau - \tau') = q_{EA}(\tau_1)f\left(\frac{\tau - \tau'}{A(\tau_1)}\right)$ with an exponential scaling function $f(x)$ for large $x$. We have checked that all the results discussed in this section are independent of the number $N$ of spins, provided it is sufficiently large.

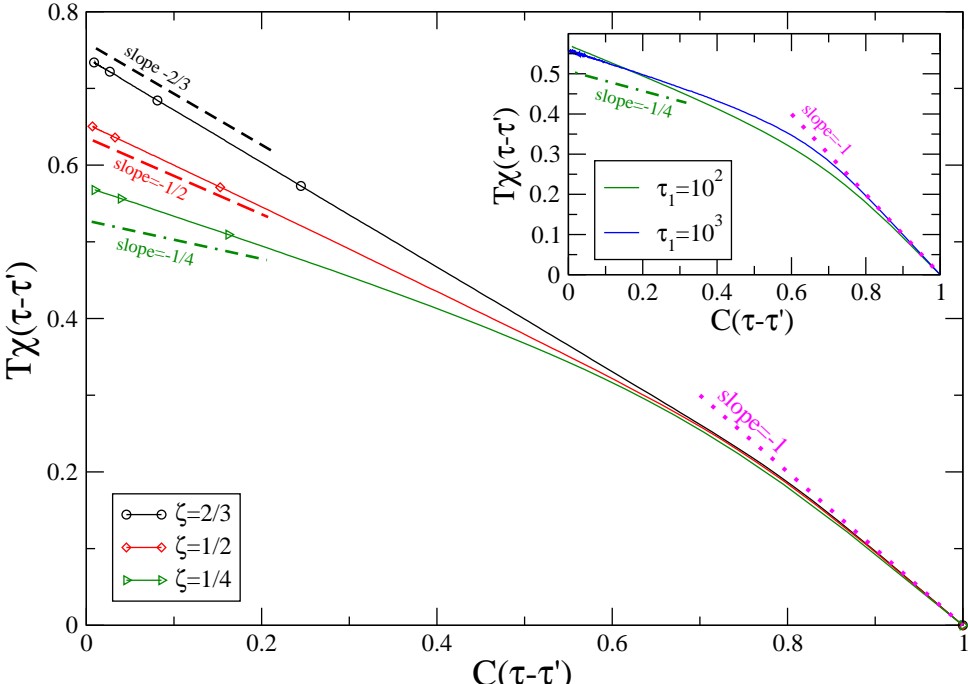

Figure 2: The fluctuation-dissipation plot of the model with multithermalized coupling constants $J_{ij}$. The integrated response $T\chi(\tau, \tau')$ is plotted against the autocorrelation function $C(\tau, \tau')$ for a system of $N = 2^{10}$ spins with random bimodal coupling constants $J_{ij} = \pm 1/\sqrt{N}$. The temperature of the spins is $T = 1/2$ and that of the $J_{ij}$s is $T' = T/\zeta$, with the three values $\zeta = 2/3$, $1/2$, and $1/4$. All the curves are computed on a stationary system, which is obtained by letting $\tau' = 50$ for $\zeta = 2/3$ and $\zeta = 1/2$, and $\tau' = 25$ for $\zeta = 1/4$. Data are averaged over $5 \cdot 10^4 - 10^6$ realisations, for the various cases. The perturbation applied for the computation of the response function is $H = 0.1$. The timescale of the slow bath is $\tau_1 = 10^2$ for all the curves (except the blue one in the inset, for which $\tau_1 = 10^3$). The dotted line is the equilibrium slope $-1$ while the dashed ones are the expected slopes in the small $C$ sector. In the inset a comparison is presented for $\zeta = 1/4$ between the system with $\tau_1 = 10^2$ and $\tau_1 = 10^3$.

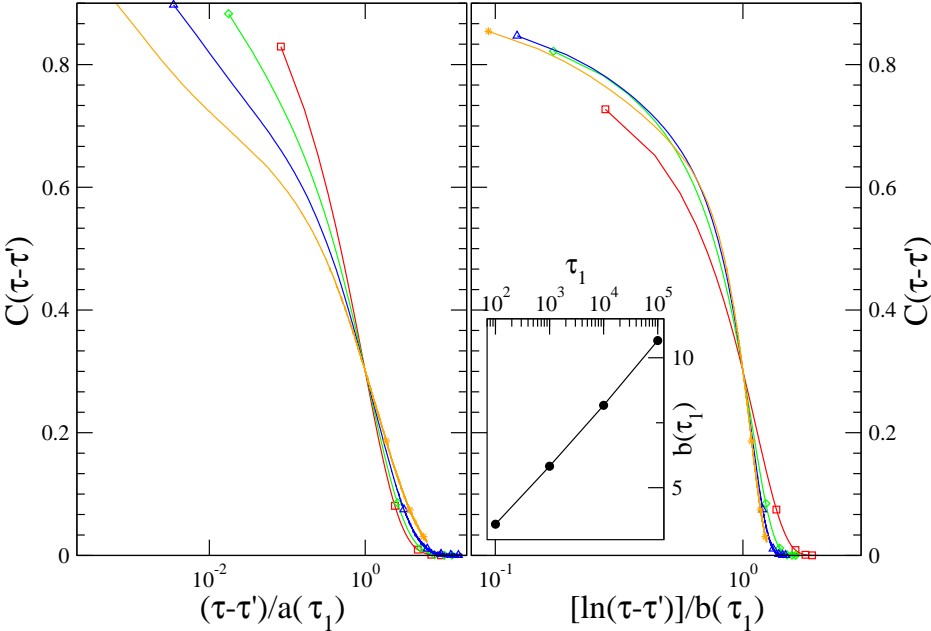

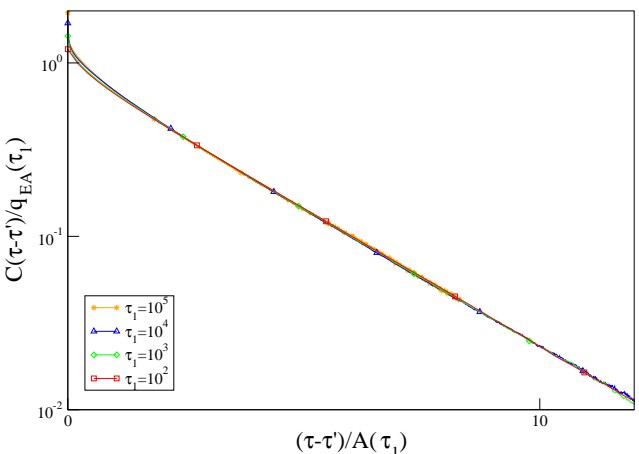

Figure 3: Three scalings for the autocorrelation function $C(\tau, \tau')$ of the SK model with $T = 1/2$ and the coupling constants $J_{ij}$ multithermalized with $T' = \infty$; and various timescales $\tau_1$ of the slow bath (see key in the lower panel). Stationarization is achieved by letting $\tau' = 25, 50, 3000, 3000$ for $\tau_1 = 20^2, 10^3, 10^4, 10^5$, respectively. Upper left panel: $C(\tau - \tau')$ is plotted against $(\tau - \tau')/a(\tau_1)$, where $a(\tau_1)$ is a fitting parameter adjusted so to collapse the curves at $C(\tau - \tau') = 0.3$. Upper right panel: $C(\tau - \tau')$ is plotted against $\ln(\tau - \tau')/b(\tau_1)$, where $b(\tau_1)$ is adjusted to collapse the curves at $C(\tau - \tau') = 0.3$ and is plotted in the inset on a log-linear plot (best fit yields $b(\tau_1) = -1.18 + 1.02 \cdot \ln \tau_1$). Bottom panel: $C(\tau - \tau')/q_{EA}(\tau_1)$ plotted against $(\tau - \tau')/A(\tau_1)$ on a linear-log scale, where $A(\tau_1)$ and $q_{EA}(\tau_1)$ are fitting parameters determined as to obtain data collapse among curves at their tails. The system size is $N = 2^{10}$, the coupling constants are random $J_{ij} = \pm 1/\sqrt{N}$. All the curves are averaged over $4 \cdot 10^4 - 2 \cdot 10^5$ realisations, for the various cases.

# 3 Stationarization and Multithermalization

## 3.1 Stationarization

Some systems never thermalize in the thermodynamic limit: their equilibration time diverges with the size. The simplest case is phase-separation starting from a mixed situation: the domains of the phases grow with time, and take a time that depends on the size of the sample to achieve their stationary situation. Another example is the case of spin-glasses, where the divergence with system size is much more rapid. There are some systems like ordinary structural glasses, of which we do not know if the equilibration time is infinite or just longer than we can measure. In all these cases, an autocorrelation function of any observable has a form that is not time-translational invariant, such as for example in domain growth where $C(\tau, \tau') = C_1(\tau - \tau') + C_2(\tau'/\tau)$ for $\tau > \tau'$. Such a situation (with $C_2 \neq 0$) is called 'aging'.

Systems of this kind may achieve stationarity once subjected to a random, time-dependent interaction [16], *even in the thermodynamic limit*. By this we mean that all time correlations and response functions, averaged over randomness, become time-translational invariant (TTI). This does *not* mean that the system is in thermal equilibrium, but rather in a non-equilibrium stationary state. A very intuitive example of this is the case of thermalized couplings discussed in section 2.4. The $J_{ij}$ evolve with a slow timescale $\tau_1$ in contact with a bath of a higher temperature than that of the spins, and follows a Monte Carlo with renewals every $\tau_\sigma$. If the fast bath has very low temperature, the spins will be 'trying to optimize" the free energy, but they will be 'chasing' a continuously changing optimum, and will not be able to improve beyond some ($\tau_\sigma$-dependent) level. The dynamics then becomes stationary with a timescale of order $\tau_1$.

A more subtle situation is that of a system subjected to a linear field , itself slowly evolving as in section 2.1. Because the low temperature configurations depend strongly on the fields $J_i$ ('chaos in field'), again one would expect that the slightest field would stationarize the evolution. However, this cannot be true for arbitrary $\tau_1$, since for $\tau_1$ small the field merely contributes to the fast bath, and amounts only to a rise in its temperature: aging does not disappear if this change is not strong enough. We thus have to expect stationarity to happen in a region of the plane $1/\gamma_1, 1/\tau_1$ around the corner $(0,0)$. We have studied numerically this plane Checking for stationarity can be an hard and ambiguous task, particularly for large $\tau_1$. In order to do that we used the following criterion: we stipulate that the system stationarizes if the energy $\langle H \rangle$ becomes time independent and/or the autocorrelation becomes stationary and/or it shows an exponential decay (indeed we noticed that the decay of $C$ as a function of $\tau - \tau'$ for fixed $\tau'$ is approximately exponential or much slower if the system is stationary or ages). The result of our studies is shown in Fig. 4 which confirms what we expected.

Finally, let us note that more complicated situations are possible. A multibath may *partially stationarize* a system which originally had autocorrelations decaying from $q_d$ to $q$ in a time-translational manner, and from $q$ to zero in an aging one, by enlarging the range of stationarity $q_d - q'$ [9]. Let us also mention here that some systems, specifically ferromagnets, typically do not stationarize when detailed balance is broken by coupling to different baths [33] or when mechanically driven [34]. The reason seems to be that their effective temperature is infinite.

## 3.2 Multithermalization with a multibath

Let us consider a multibath with two-time correlations $\mathcal{C}$, response $\mathcal{R}$ and integrated response $\tilde{\chi}(\tau) = \int_0^\tau \mathcal{R}(\tau')d\tau'$ and a stationary system with two-time correlations $C$ and response $R$ and integrated response $\chi(\tau) = \int_0^\tau R(\tau')d\tau'$. For example one may think of the multibath as realized with external fields and the stationary system being a pure SK model as in section 2.3.

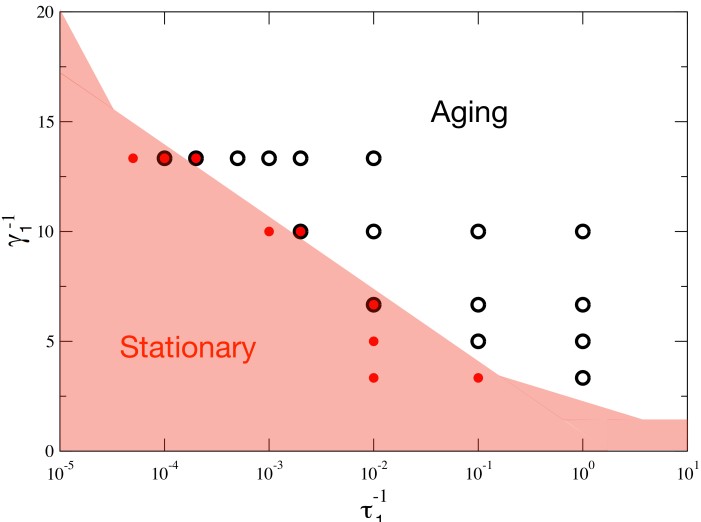

Figure 4: The *phase-diagram* of the model with a multithermalized external field. The system is made of $N = 2^{12}$ spins with random bimodal coupling constants $J_{ij} = \pm 1/\sqrt{N}$. The spin temperature is $T = 0.7$ and the field temperature is $T' = \infty$. The points in the figure represent all the parameter choices investigated. The black circles correspond to an aging system, the red dots to a stationary one. Black circles with a red dot inside correspond to ambiguous cases where the three criteria discussed in the text give weak or opposite indications. The shaded red region is a pictorial representation of the stationary phase.

At each time scale we consider the effective temperatures (40)

$$\frac{1}{\tilde{T}(z)} = \tilde{\beta}(z) = -\frac{d\tilde{\chi}}{d\mathcal{C}} \qquad ; \qquad \frac{1}{T(z)} = \beta(z) = -\frac{d\chi}{dC}, \qquad (53)$$

where $z = \tau - \tau'$ is the time difference. We say that the multibath and the stationary system are *multithermalized* if at each $z$ the temperatures are the same $T(z) = \tilde{T}(z)$, for all the pair of observables of the system used to define $C, R$. In other words, we have a multi-fluctuation-dissipation relation consistent with that of the bath (32). We shall see in section 5 that this has strong implications for then equilibrium measure.

Let us anticipate when we expect multithermalization to happen. *i)* Any system with short timescales in contact with a multibath (with suitably separated timescales) develops the scales that are thermalized with those of the multibath. *ii)* Systems that do not become stationary (they age) in contact with an ordinary bath, may become stationary in contact with a multibath. *iii)* However, only if the multibath's temperatures coincide with the natural aging temperatures of the system multithermalization may be achieved with minimal energy transport, as we shall see in the dynamical version of Guerra's scheme in Sec. 7. The possibility that the system synchronizes its timescales with those of the bath so as to make temperatures match, exists if the system has reparametrization invariances [7].

### 3.3 Work and power of a multibath

Let us compute the work per unit time (power) after the classical definition $W =$ force $\times$ velocity. Using the quantities introduced in section 2.2 and (26) we obtain

$$W = \left\langle \sum_{a,i} \gamma_a J_i^a(\tau)\, \dot{\sigma}_i(\tau) \right\rangle_{dyn} = \sum_{a,i} \gamma_a \left\langle \hat{\rho}_i^a\, \dot{\sigma}_i \right\rangle_{dyn} + \gamma_a^2 \int^\tau d\tau'\, \mathcal{R}_a(\tau - \tau') \left\langle \dot{\sigma}(\tau)\sigma(\tau') \right\rangle_{dyn}. \tag{54}$$

The average in (54) is computed over the dynamics (i.e. over the noises). Since the noise is Gaussian one can rewrites the terms $\left\langle \rho_i^a \dot{\sigma}_i \right\rangle_{dyn}$ using integration by parts. Let us start recalling that the evolution of $\sigma$ follows (15a):

$$\text{Eqn}(\sigma_i) = \tau_\sigma \dot{\sigma}_i - \gamma \sum_j J_{ij}\sigma_j - \sum_a \gamma_a J_i^a - \eta_i = 0 \quad \text{with} \quad \langle \eta_i(\tau)\eta_j(\tau') \rangle = 2\tau_\sigma T \delta_{ij}\delta(\tau - \tau'). \tag{55}$$

Rewriting the solutions of (55) using the Fourier representation of the Dirac delta

$$\int d\hat{\sigma} e^{\int d\tau' \sum_i \hat{\sigma}_i(\tau')[\text{Eqn}(\sigma_i)]}, \tag{56}$$

and keeping track only on the $J_i^a$ term one obtains for the first term of the r.h.s. of (54) the expression

$$\left\langle \hat{\rho}_i^a \dot{\sigma}_i \right\rangle_{dyn} = \int d\sigma d\hat{\sigma} \left\langle e^{\int d\tau' \sum_i \hat{\sigma}_i(\tau')[\sum_a \gamma_a J_i^a(\tau') + \text{other terms}]} \hat{\rho}_i^a \dot{\sigma}_i \right\rangle_{dyn}. \tag{57}$$

Now one uses (26) to rewrite the above quantity as

$$\int d\sigma d\hat{\sigma} \left\langle e^{\int d\tau' \sum_i \hat{\sigma}_i(\tau')[\sum_a \gamma_a \hat{\rho}_i^a(\tau') + \text{other terms}]} \hat{\rho}_i^a \dot{\sigma}_i \right\rangle_{dyn}, \tag{58}$$

where the other terms in the exponent does not depend on the Gaussian noise $\rho_i^a$. The covariance of $\rho_i^a$ is given by (28), hence integration by parts leads to

$$\left\langle \hat{\rho}_i^a \dot{\sigma}_i \right\rangle_{dyn} = \gamma_a \int d\tau' \mathcal{C}_a(\tau - \tau') \left\langle \dot{\sigma}_i(\tau)\hat{\sigma}_i(\tau') \right\rangle_{dyn}. \tag{59}$$

Now collecting all terms and keeping in mind the definitions of $\mathcal{R}, \mathcal{C}$ given in (30) and (31), we get

$$W = \int^\tau d\tau'\, \partial_\tau R(\tau - \tau')\mathcal{C}(\tau - \tau') + \int^\tau d\tau'\, \partial_\tau C(\tau - \tau')\mathcal{R}(\tau - \tau'), \tag{60}$$

where

$$C(\tau, \tau') = \sum_i \langle \sigma_i(\tau)\sigma_i(\tau') \rangle_{dyn} \quad ; \quad R(\tau, \tau') = \sum_i \langle \sigma_i(\tau)\hat{\sigma}_i(\tau') \rangle_{dyn} \tag{61}$$

are the correlation and response functions of the system. We may obtain a more explicit expression by integrating by parts in time, and introducing the (time-dependent) effective temperatures

$$\mathcal{R}(z) = \tilde{\beta}(z)\partial_z \mathcal{C}(z) \quad and \quad R(z) = \beta(z)\partial_z C(z), \tag{62}$$

so that (60) becomes

$$W = \int^\tau d\tau'\, \partial_\tau C(\tau - \tau')\partial_\tau \mathcal{C}(\tau - \tau')\left[\beta - \tilde{\beta}\right](\tau - \tau'). \tag{63}$$

This indeed looks like a conduction term. Since $\mathcal{C} = \sum_a \gamma_a^2 \mathcal{C}_a$, we may discriminate the work done at each timescale writing $W = \sum_a W_a$ with

$$W_a = \gamma_a^2 \int^\tau d\tau' \, \partial_\tau C(\tau - \tau') \partial_\tau \mathcal{C}_a(\tau - \tau') [\beta_a - \tilde{\beta}_a], \tag{64}$$

where we used the fact the $[\beta - \tilde{\beta}](\tau - \tau')$ is constant within each timescale. This is an energy per unit time. If we adimensionalize separately every timescale, we get

$$\tau_a W_a = \gamma_a^2 \int^\tau d\tau' \, \partial_\tau C(\tau - \tau') \mathcal{C}_a(\tau - \tau') [\beta_a - \tilde{\beta}_a]. \tag{65}$$

This is energy transferred by the $a$-bath during a time $\tau_a$, and they all vanish upon the multi-thermalization condition $\beta \equiv \tilde{\beta}$ .

# 4 Large-time limits.

At this point we need to be more precise about what we mean by 'large times' and their structure. If the system is finite, the answer is straightforward: we need that at each timescale the system has had enough time to reach the final distribution of all the 'faster' variables, while the 'slower' ones are still substantially unchanged. All this fixes a hierarchy. What about systems in the thermodynamic limit $N \to \infty$ ?

## 4.1 Mean-field systems

Consider the paradigmatic case of the Langevin dynamics on a $p$-spin glass with energy $\sum_{i_1 \dots i_p} J_{i_1,\dots,i_p} \sigma_{i_1} \cdots \sigma_{i_p}$ for $p > 2$. This example is interesting because it shows us where things can go differently in the thermodynamic limit. The *complexity* landscape [20] is constituted as in Fig. 5, the states have a density $e^{N\Sigma(f)}$ and stop abruptly at a 'threshold level'. In times of order one, the dynamics age – without becoming stationary – just over the threshold energy of the highest, and overwhelmingly more numerous, states. At times $t \sim e^{KN}$ the dynamics penetrate down to a $K$-dependent free energy density below the threshold level. Note that each 'step down' in free energy density takes an exponentially (in $N$) time longer than the previous one. At each free-energy, we define the effective temperature $\frac{1}{T_{eff}} = \frac{\partial \Sigma(f)}{\partial f}$ as in Fig. 5.

Let us now couple the system to a slow bath of intensity $\gamma_1$, temperature $T_1$ and timescale $\tau_1$. This is the dynamic counterpart of a well-studied procedure, see [30] (see also [29] and [31]). For very long $\tau_1$, longer than any timescale of a finite system, and small $\gamma_1$ (again, but not vanishing with $N$), the system multithermalizes and we find that it eventually sticks at a level with $T_{eff} = T_1$. Starting from a high energy situation, this takes a long time: the system has to age its way down to the appropriate multi-equilibrium level, and this takes exponentially long in $N$.

If we consider times of order one with the thermodynamic limit taken first, the system becomes stationary for an arbitrarily weak bath provided its temperature is $T_1 \geq T_{eff}^{threshold}$. It may furthermore multithermalize in times that are large but still do not diverge with $N$, but if and only if $T_1 = T_{eff}^{threshold}$ . We have hence discovered that if the thermodynamic limit is taken first, in this case a stationary situation exists with energy densities higher than the equilibrium one, even with a field of low amplitude and timescale of order one. This, we shall claim, cannot happen in finite dimensions.

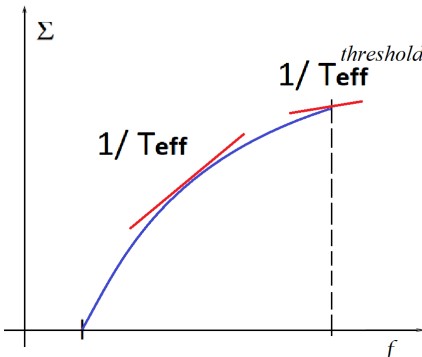

Figure 5: Schematic picture of the complexity $\Sigma$ as a function of the free energy $f$ for the $p$-spin glass model. The dotted line represents the threshold free energy. The red line is the slope of the function $\Sigma(f)$ and determines the effective temperature of the system

## 4.2   Finite-dimensional systems

In a remarkable paper, Franz et al [19] argued that by using the property of stochastic stability introduced in [1] the quantities calculated out of equilibrium in an aging *finite dimensional system with short-range interactions* at long times coincide with the ones at equilibrium. For a rigorous study of stochastic stability and its consequences in finite dimensional systems see [2, 3]. In particular, this is expected (see [35] for a discussion of this hypothesis) for the susceptibilities [8] associated with a perturbation $H \rightarrow H + \epsilon \sum_{i_1,...,i_r} J_{i_1,...,i_r} \sigma_{i_1}...\sigma_{i_r}$:

$$I^{(r)} = \frac{\partial}{\partial \epsilon} \left\langle \sum_{i_1,...,i_r} J_{i_1,...,i_r} \sigma_{i_1}...\sigma_{i_r} \right\rangle \Bigg|_{\epsilon=0}. \tag{66}$$

The essence of their argument is the following: if we are guaranteed that the dynamics lead in times of order one (finite as $N \rightarrow \infty$) to an energy density that coincides with the one of the statics, and this for any field $J_{i_1,...,i_r}$, then, by simple derivatives of the target values we obtain the same susceptibilities in an aging system as in an equilibrated one. Hence, static and dynamic susceptibilities coincide at long (but finite in the thermodynamic limit) times. To show the convergence, they use a nucleation argument that shows that metastable states with higher free-energy densities are not possible in finite dimensions. What is important to us here is that under basically the same assumptions, we may apply their arguments to a system under the action of a multibath, to show that the timescale-separations required for the multibath, though large, need not diverge in the thermodynamic limit, if the system is finite-dimensional. For example a term like

$$\left\langle \sum_{a,i} \gamma_a J_i^a \sigma_i \right\rangle \tag{67}$$

will have the time to relax in finite times to its asymptotic value ($\tau_a \rightarrow \infty$), provided that $\gamma_a$ is of order one (also an assumption related to stochastic stability [19]). In other words, the situation we met above for the mean-field $p$-spin model ($p > 2$), where one needs $\tau_\sigma$ that are exponentially long in $N$ to reach true stationarity, cannot happen in finite dimensions.

# 5 Multi-reversible transformations, thermodynamic integration and measure

The appearance of effective temperatures in aging glassy systems, even in the absence of a multibath, has long been known [12,17]. The real power of the multi-bath appears when we use the fact of multi-thermalization with a slowly evolving bath. This allows us to infer the underlying probability distribution of a system, even in a numerical simulation of a realistic system, as we shall see in Section 8.

## 5.1 Ordinary reversible transformations, thermodynamic integration and Fluctuation-Dissipation relation

Consider a system which depends upon variables that we shall denote collectively by $\sigma$. Given two observables $A, B$ the correlation function, denoting by $\langle \cdot \rangle^{dyn}$ the dynamic average, is:

$$C_{AB}(\tau, \tau') = \langle A(\tau)B(\tau') \rangle^{dyn}. \tag{68}$$

Given a perturbation of the type $H \to H - \delta h(\tau)B$ the response function is

$$R_{AB}(\tau, \tau') = \frac{\delta \langle A(\tau) \rangle^{dyn}_{\delta h}}{\delta h(\tau')} \bigg|_{h=0}, \tag{69}$$

where $\langle \cdot \rangle^{dyn}_{\delta h}$ is the average under perturbation.

We shall consider transformations of the energy function (via its parameters) that are *reversible*, by which we mean that:

- they are quasi-static: if at any step of the transformation we were to stop and wait until the average values of all observables (over time-windows, or over several copies of the system following the same protocol) does not evolve, and, furthermore, two-time correlations depend exclusively on time-differences.

- the Fluctuation-Dissipation relation holds at each step:

$$R_{AB}(\tau, \tau') = \beta \frac{\partial}{\partial \tau'} C_{AB}(\tau, \tau'). \tag{70}$$

In practice, this means that duplicating all times involved does not change the result. Let us show that this process leads to the Boltzmann-Gibbs distribution. The idea is to consider a perturbation $\frac{\delta\beta}{\beta}H$, so the expectations will be those associated with the measure $e^{-\beta\left[H+\frac{\delta\beta}{\beta}H\right]}$ or equivalently $\beta \to \beta + \delta\beta$. Consider an arbitrary observable $A$. Then by definition of the response function (69) and by FDT (70) we have that

$$\delta \langle A \rangle^{dyn}_{\delta\beta} = -\frac{\delta\beta}{\beta} \int_{-\infty}^{\tau} R_{AH}(\tau, \tau')d\tau'$$
$$= -\delta\beta[C_{AH}(\tau, \tau) - C_{AH}(\tau, -\infty)] = -\delta\beta(\langle AH \rangle^{dyn} - \langle A \rangle^{dyn}\langle H \rangle^{dyn}), \tag{71}$$

where we have used the clustering property at widely separate times. Because this holds for every $A$, one can determine $\mu(\sigma)$, the distribution of $\sigma$. Indeed we can choose $A = \delta(\sigma - \sigma')$ and take the average over $\sigma'$ obtaining

$$\frac{d\mu(\sigma)}{d\beta} = \frac{d}{d\beta} \langle \delta(\sigma - \sigma') \rangle^{dyn} = -H(\sigma)\mu^{dyn}(\sigma) + \langle H \rangle_{\beta}\,\mu(\sigma) \quad \to$$

$$\log\mu(\sigma) = -\beta H(\sigma) + \int^{\beta} d\beta' \langle H \rangle^{dyn} \to$$

$$\mu(\sigma) = \mathcal{N}(\beta)e^{-\beta H(\sigma)}.$$

Imposing the normalization, this gives the Gibbs-Boltzmann measure. What we have done is to reconstruct this measure by means of a 'thermodynamic integration' in $\beta$.

## 5.2 Multi-reversible transformations, thermodynamic integration and multi-Fluctuation-Dissipation relation

Suppose now we have a system in contact with a multibath with sufficiently separated timescales. We consider transformations that change the Hamiltonian slowly enough, so that the transformation is *multireversible*, namely

- it is quasi-static: if at any step of the transformation we were to stop and wait, with the multibath still on, the average values of all observables (over time-windows, or over several copies of the system following the same protocol) would not evolve, and, furthermore, two-time correlations and response functions depend exclusively on time-differences.

- the Fluctuation-Dissipation Ratio of all the observables of the system and of the multibath coincide at each timescale with a single $\beta(\tau)$:

$$R_{AB}(\tau - \tau') = \beta(\tau - \tau')\frac{\partial}{\partial \tau'}C_{AB}(\tau - \tau'). \tag{72}$$

This is just the 'multi' version of ordinary reversibility. We shall show, with a procedure that is a direct generalization of the one above, that we obtain the multibath measure introduced in section 2.

Let us assume that we evolve $T = 1/\beta$, the fast temperature, from $T = 0$ to any finite temperature. Consider a system with fast $\sigma$ and slow $J$ variables at inverse temperature $\beta_1$ and relaxation time $\tau_1$. We shall assume multi (bi) thermalization namely with effective temperature $\beta(\tau - \tau') = \beta x(\tau - \tau')$ where

$$x(\tau - \tau') = \begin{cases} 1 & if \ \tau - \tau' < \tau^* \\ \zeta = \frac{\beta_1}{\beta} & if \ \tau - \tau' > \tau^* \end{cases}, \tag{73}$$

for some $\tau^*$ such that $1 \ll \tau^* \ll \tau_1$. Hence we are assuming that the system spontaneously respects (72), for all observables $A, B$, namely FDT with temperatures $\beta$ in the fast timescales, and $\beta_1$ in the slow timescales. This corresponds to $\mu(\sigma|J)$ and $\mu(J)$, the former being the distribution reached by $\sigma$ before $J$ had the time to move. These are the distributions we wish to compute. We proceed as above, treating energy as a perturbation, but this time the r.h.s. of (71) can be split into timescales. We choose the time $\tau^*$ such that $\sigma$ has performed all its fast relaxation, but $J$ has not had the time to change. Linear response (69) reads as

$$
\begin{aligned}
-\frac{\delta \langle A\rangle_{\delta\beta}^{dyn}}{\delta\beta} &= \frac{1}{\beta}\int_{-\infty}^{\tau} R_{AH}(\tau, \tau')d\tau' = \frac{1}{\beta}\int_{\tau^*}^{\tau} R_{AH}(\tau, \tau')d\tau' + \frac{1}{\beta}\int_{-\infty}^{\tau^*} R_{AH}(\tau, \tau') \\
&= \langle AH\rangle^{dyn} - (1-\zeta)\langle A(\tau)H(\tau^*)\rangle^{dyn} - \zeta\langle A\rangle^{dyn}\langle H\rangle^{dyn}.
\end{aligned} \tag{74}
$$

Now, we make the crucial assumption that the time-difference $\tau - \tau^*$ is large enough that $\sigma$ is able to thermalize at given $J$, yet small enough that $J$ hasn't changed. Then, the expectation

$\langle A(\tau)H(\tau^*)\rangle^{dyn}$ partially clusters, and (74) corresponds to

$$
\begin{aligned}
\frac{\delta\langle A\rangle}{\delta\beta} =& -\int d\sigma dJ\, AH\, \mu(\sigma|J)\,\mu(J) \\
&+(1-\zeta)\int dJ\mu(J)\left(\int d\sigma' A\mu(\sigma'|J)\right)\left(\int d\sigma'' H\mu(\sigma''/J)\right) \\
&+\zeta\left(\int d\sigma' dJ'\, A\,\mu(\sigma'|J')\,\mu(J')\right)\left(\int d\sigma'' dJ''\, H\,\mu(\sigma''|J'')\,\mu(J'')\right).
\end{aligned} \tag{75}
$$

Choosing $A(\sigma',J')=\delta(\sigma-\sigma')\delta(J-J')$ we get

$$
\frac{d}{d\beta}\left(\mu(\sigma|J)\mu(J)\right)=\left\{-H(\sigma,J)+(1-\zeta)\left[\int d\sigma' H(\sigma',J)\mu(\sigma'/J)\right]+\zeta r(\beta)\right\}\mu(\sigma|J)\,\mu(J), \tag{76}
$$

where $r(\beta)\equiv\int d\sigma dJ\, H\,\mu(\sigma/J)\,\mu(J)$. Integrating once over $\sigma$, and rearranging, we get the two equations:

$$
\begin{aligned}
\frac{d}{d\beta}\log[\mu(\sigma|J)\mu(J)] &= -H(\sigma,J)+(1-\zeta)\left[\int d\sigma' H(\sigma',J)\mu(\sigma'/J)\right]+\zeta r(\beta) \\
\frac{d}{d\beta}\log\mu(J) &= -\zeta\left[\int dx\sigma' H(\sigma',J)\mu(\sigma'/J)\right]+\zeta r(\beta),
\end{aligned} \tag{77}
$$

and subtracting them we find

$$
\begin{aligned}
\frac{\delta\log\mu(\sigma|J)}{\delta\beta} &= -H(\sigma,J)+\left[\int d\sigma' H(\sigma',J)\mu(\sigma'|J)\right] \\
\frac{\delta\log\mu(J)}{\delta\beta} &= -\zeta\left[\int d\sigma' H(\sigma',J)\mu(\sigma'/J)\right]+\zeta r'(\beta).
\end{aligned} \tag{78}
$$

The solution of the first equation is $\mu(\sigma|J)=g(J,\beta)e^{-\beta H(\sigma,J)}$ for some $g(J,\beta)$ that may be fixed by normalization:

$$
\mu(\sigma|J)=\frac{e^{-\beta H(\sigma,J)}}{\int d\sigma'\, e^{-\beta H(\sigma',J)}}.
$$

Plugging this into the second equation, it becomes:

$$
\begin{aligned}
\frac{\delta\log\mu(J)}{\delta\beta} &= -\zeta\left[\int d\sigma' H(\sigma,J)\frac{e^{-\beta H(\sigma,J)}}{\int d\sigma'\, e^{-\beta H(\sigma',J)}}\right]+\zeta r'(\beta) \\
&= \zeta\frac{\delta}{\delta\beta}\ln\left[\int d\sigma'\, e^{-\beta H(\sigma',J)}\right]+\zeta r'(\beta),
\end{aligned} \tag{79}
$$

which implies:

$$
\mu(J)=\frac{\left[\int d\sigma' e^{-\beta H(\sigma',J)}\right]^{\zeta}}{\int dJ\left[\int d\sigma' e^{-\beta H(\sigma',J)}\right]^{\zeta}}, \tag{80}
$$

which matches (22). The generalization to $r$ nested timescales is straightforward, one must be able to choose times $\tau_1^*,\dots,\tau_r^*$ such that at $\tau-\tau_a^*$ the variables $J_1,\dots,J_a$ did not have the time to move, while $J_{a+1},\dots,J_r,\sigma$ have reached their equilibrium distribution. The effective temperature is a staircase function taking values $\beta\zeta_a$ for $a=1\dots,r$.

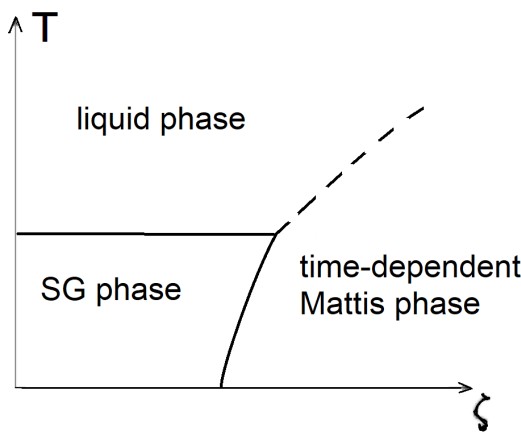

Figure 6: A sketch of the $\zeta$-$T$ phase diagram.

# 6 Fluctuating couplings and a physical vision of $T-\zeta$ phase-diagram

As mentioned above, the average (5) was originally only motivated by the value around $\zeta = 0$. At $\zeta = 1$ we have the annealed average, and it is easy to check that there is no transition at any temperature for a spin-glass. Clearly, at low enough temperatures there is a spin-glass phase as $\zeta \to 0$. The question then arose of what is the transition line in the $\zeta - T$ plane: this was computed for the random energy model by Gardner and Derrida [18] and later by Guerra and Talagrand by rigorous methods [24, 26], for the SK model by Kondor [15] using replica approach (see [4, 26] for a rigorous treatment). In both cases, the phase diagram looks like the sketch in Fig 6.

This may be obtained by a two-temperature multibath, where the spins are at temperature $T$ and the couplings at temperature $T' = T/\zeta$, and evolve at a much slower rate $\tau_1$ as explained in section 2.4. The interpretation of the phase diagram is physically appealing, and may be seen in the sketch of Figure 7 and 8. The values of effective temperatures $\beta x$ of the system are cut off at the level of $T' = T/\zeta$ as follows: the slope of the curve $\chi$ vs $C$ is approximately independent of $\zeta$ from $q_{EA}$ down to the point $q_{min}$ where the tangent $\frac{d\chi}{dC} = -\frac{\zeta}{T}$; and continues as a straight line down to the minimal value of $C$ (see Fig. 7). Starting from zero and increasing $\zeta$, the transition takes place at the value of $\zeta = x_{crit}$ (at which $T'$ matches the lowest available

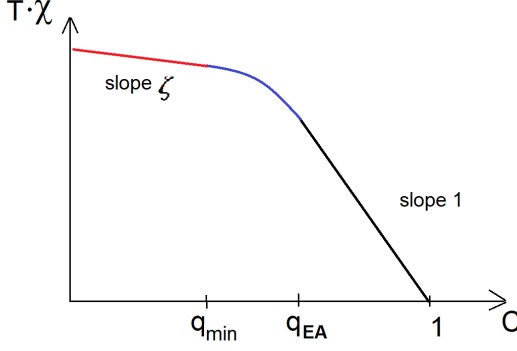

Figure 7: The $\chi$ vs. $C$ curve for a system with couplings at temperature $\frac{T}{\zeta}$. The curved part of the plot is almost independent of temperature, and the straight part matches tangentially the curved part. The transition in $\zeta$ takes place at the point where this tangent happens at the largest value $C < q_{EA}$

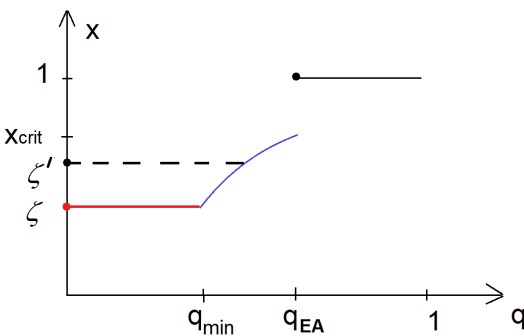

Figure 8: Here the derivative of the function $-T\chi(C)$ : the curved part is essentially independent of the temperature, while the plateau to the left is given by $\zeta$. The transition takes place when the line $x = \zeta$ intersects the curved part

effective temperature for the system): at this point the relaxation takes place with only two possible values for $x(q)$ (see Fig. 8). At the transition point one should observe in Fig. 7 two straight lines with slope $-\zeta$ and $-1$, that intersect with each other at the point $C = q_{EA}$.

Within the two-temperature interpretation of this diagram, it seems possible to make a phenomenological description in terms of droplets within the glass phase of this diagram, and this is a framework that would render the different approaches directly comparable.

# 7 Physically implementing Guerra's interpolation as a multireversible transformation

The multibath measure generated by (13) is the core of the Guerra's interpolation scheme for the SK model [14]. In section 2.1 we showed that a multibath measure can be viewed as a stationary measure for a dynamical system in contact with different thermal baths and widely separated timescales, hence it is natural to look for a dynamical analogous of Guerra interpolation. In this section we investigate this analogy by addressing in particular the following question: is there a dynamical counterpart of the positivity property in Guerra's scheme? We will show that if the system *multithermalizes* (see section 3.2 for the precise meaning) along the interpolating path then the answer to the previous question is positive thanks to the property of a *multi-reversible transformation* (described in section 5.2).

Let us start by briefly sketching Guerra's construction, we refer to the original work for the details [14]. Consider a system of $N$ spins and two independent random Hamiltonian $H(\sigma)$ and $\tilde{H}(\sigma)$ with centered gaussian disorder and covariances

$$\mathbb{E}\big[H(\sigma^1)H(\sigma^2)\big] = \frac{N}{2}(q_{12})^2 \quad \text{and} \quad \mathbb{E}\big[\tilde{H}(\sigma^1)\tilde{H}(\sigma^2)\big] = N q_{12}, \tag{81}$$

where $q_{12} = \frac{1}{N}\sum_{i \leq N}\sigma_i^1\sigma_i^2$ is the overlap. In other words $H$ is the Hamiltonian of the SK model while $\tilde{H}$ is a gaussian external field. Let $t \in (0,1)$ be an interpolation parameter and $r$ be an integer. Consider a non decreasing sequence $q = (q_a)_{a \leq r}$ with $q_0 = 0, q_r = 1$. Let $(H^a)_{1 \leq a \leq r}$ be a family of i.i.d. copies of $\tilde{H}$ and define

$$\mathcal{H}(\sigma) = \sum_a \sqrt{q_a - q_{a-1}}\, H^a(\sigma), \tag{82}$$

and for $t \in (0,1)$ the interpolating Hamiltonian

$$H_t(\sigma) = \sqrt{t}H(\sigma) + \sqrt{1-t}\,\mathcal{H}(\sigma). \tag{83}$$

Then $H_1$ is the Hamiltonian of the SK model while $H_0$ contains just one-body interactions. Following Guerra, for $H_t$ we assume a multibath measure associated to a given non decreasing sequence $\zeta = (\zeta_a)_{a \leq r}$. The generating functional or pressure density for this measure is

$$p(t) = \frac{1}{N} \mathbb{E} P_0(t), \tag{84}$$

where $\mathbb{E}$ averages the quenched variables and $P_0(t)$ is obtained trough the recursion (13) starting with

$$P_r(t) = \ln Z_r(t) = \log \sum_{\sigma} e^{-\beta H_t(\sigma)}. \tag{85}$$

We denote by $\langle\ \rangle_t$ the average w.r.t. the multibath measure induced by (84) and $x(q)$ denotes the discrete distribution associated to the sequences $q$ and $\zeta$. Then one can prove a crucial inequality

$$\frac{d}{dt} p(t) = -\frac{\beta}{N} \left\langle \frac{dH_t}{dt} \right\rangle_t \leq -\frac{\beta^2}{2} \int_0^1 q\, x(q)\, dq, \tag{86}$$

for any choice of the sequences $q$ and $\zeta$. Integrating both sides of (86) from $t = 0$ to $t = 1$ on gets the celebrated Guerra's Replica Symmetry Broken bound [14]. Notice that this procedure can be viewed as a *thermodynamic integration*. Next we will show how one can obtain an inequality analogous to (86) in the dynamical setting.

## 7.1 Dynamic realization

Consider a dynamical system with Hamiltonian (14) and set $\gamma = \sqrt{\frac{t}{N}}$ and $\gamma_a = \sqrt{1-t} \sqrt{q_a - q_{a-1}}$ for $a \leq r$. We assume as before that the quenched couplings $J_{ij}$ are *i.i.d.* standard gaussian. We write the Hamiltonian as $H_t = H_0 + H_1 + H_2$:

$$
\begin{aligned}
H_0 &= -\sqrt{\frac{t}{N}} \sum_{i,j} J_{ij} \sigma_i \sigma_j - \sqrt{1-t} \sum_a \sqrt{q_a - q_{a-1}} \sum_i J_i^a \sigma_i, \\
H_1 &= A \sum_i (\sigma_i^2 - 1)^2, \\
H_2 &= \sum_{ai} \frac{1}{2} (J_i^a)^2,
\end{aligned}
\tag{87}
$$

where $A$ is a large constant forcing $\sigma_i^2 \sim 1$, so that $H_1$ is for large $A$ essentially a constant. We recall that $H_t$ is Guerra's Hamiltonian (83) apart from the term $H_1$ that allows us to use Langevin for $\sigma_i$.

The evolution of $\sigma$ follows, see eq. (15a),

$$\dot{\sigma}_i = \gamma \sum_j J_{ij} \sigma_j + \sum_a \gamma_a J_i^a + \eta_i \quad \text{with} \quad \langle \eta_i(\tau) \eta_j(\tau') \rangle = 2T \delta_{ij} \delta(\tau - \tau'), \tag{88}$$

where we have chosen $\tau_\sigma = 1$ and $T$ is the temperature of the main bath. The $J$ evolve according to (15b):

$$\tau_a \dot{J}_i^a = -J_i^a + \gamma_a \sigma_i + \rho_i^a(\tau) \ ; \ \langle \rho_i^a(\tau) \rho_j^b(\tau') \rangle = 2T^a \tau_a \delta_{ij} \delta_{ab} \delta(\tau - \tau'). \tag{89}$$

Consider now the quantity

$$-\frac{\partial H_t}{\partial t} = \frac{1}{2\sqrt{tN}} \sum_{i,j} J_{ij} \sigma_i \sigma_j - \frac{1}{2\sqrt{(1-t)}} \sum_a \sqrt{q_a - q_{a-1}} \sum_i J_i^a \sigma_i. \tag{90}$$

In complete analogy with Guerra's method, we wish to compute the average of (90) over the dynamics (i.e. over the noises). We shall need the measure for this. For the fields $J_i^a$ it is simply the Wiener measure

$$\Pi_a e^{-\frac{1}{2T_a \tau_a} \int d\tau \, \sum_i \left( \tau^a j_i^a + \frac{\partial H}{\partial J_i^a} \right)^2},$$  (91)

while for the $\sigma$ we complete the measure with a Fourier representation of Dirac-$\delta$ for equation (88):

$$\int d\hat{\sigma} \, e^{\left( \int d\tau' \sum_{ia} \gamma_a \hat{\sigma}_i(\tau') J_i^a(\tau') + \gamma \sum_{ij} J_{ij} \hat{\sigma}_i(\tau') \sigma_j(\tau') \dots + \text{other terms} \right)},$$  (92)

where we have only specified the terms containing $J_i^a$ and $J_{ij}$. We denote by $\langle \, \rangle_{dyn}$ the average with respect to the measure

$$\mathbb{E} \int d\sigma \, d\hat{\sigma} \, dJ \, e^{-\int d\tau' S(\sigma, \hat{\sigma}, J) + \text{other terms}},$$  (93)

where $\mathbb{E}$ is the average on the quenched variables and

$$S(\sigma, \hat{\sigma}, J) = \sum_{i,a} \frac{1}{2T_a \tau_a} \left( \tau_a j_i^a + J_i^a - \gamma_a \sigma_i \right)^2 - \sum_i \hat{\sigma}_i \left( \gamma \sum_j J_{ij} \sigma_j + \sum_a \gamma_a J_i^a \sigma_i \right).$$  (94)

## 7.2 Integrations by parts

We start noticing that $\left\langle -\frac{\partial H_t}{\partial_t} \right\rangle_{dyn}$ is the average of a sum of terms containing the variables $J_{ij}$ and $J_i^a$. We will rewrite those terms using integration by parts.

### 7.2.1 The $J_{ij}$

The variables $J_{ij}$ are quenched then can use integration by parts for Gaussian vectors. Hence

$$\left\langle \sum_{ij} J_{ij} \sigma_i(\tau) \sigma_j(\tau) \right\rangle_{dyn} = \sum_{ij} \mathbb{E} \int d\sigma \, d\hat{\sigma} \, dJ \, e^{\gamma \int^\tau d\tau' J_{ij} \hat{\sigma}_i \sigma_j(\tau') + \dots} J_{ij} \sigma_i(\tau) \sigma_j(\tau),$$  (95)

where we have used (88) and (92). Integration by parts in this case means to replace in the average $J_{ij} \to \frac{\partial}{\partial J_{ij}}$, and then we get:

$$\sqrt{\frac{1}{tN}} \left\langle \sum_{i,j} J_{ij} \sigma_i(\tau) \sigma_j(\tau) \right\rangle_{dyn} = \frac{1}{N} \sum_{i,j} \int^\tau d\tau' \left\langle \sigma_i(\tau) \sigma_j(\tau) \sigma_i(\tau') \hat{\sigma}_j(\tau') \right\rangle_{dyn}.$$  (96)

There are two interpretations of this term. The most general one is to notice that this is a response to fields acting on products $\sigma_i \sigma_j$ in the links that are coupled. That is, we add a term in the energy $\sum_{ij} h_{ij} \sigma_i \sigma_j$ and compute $\sum_{ij} \frac{\delta h_{ij}}{\delta h_{ij}} \langle \sigma_i \sigma_j \rangle_{h=0}$. Using the fact that correlations among sites vanishes, we rewrite the last term in (96) as

$$\frac{1}{N} \sum_{i,j} \int^\tau d\tau' \langle \sigma_i(\tau) \sigma_j(\tau) \sigma_i(\tau') \hat{\sigma}_j(\tau') \rangle_{dyn} = N \int^\tau d\tau' C(\tau, \tau') R(\tau, \tau'),$$  (97)

where

$$C(\tau, \tau') = \frac{1}{N} \sum_i \langle \sigma_i(\tau) \sigma_i(\tau') \rangle_{dyn} \quad ; \quad R(\tau, \tau') = \frac{1}{N} \sum_i \langle \sigma_i(\tau) \hat{\sigma}_i(\tau') \rangle_{dyn},$$  (98)

are the correlation and response functions of the system to the perturbation $h_{ij}$.

### 7.2.2 The $J_i^a$

The core of Guerra's construction is the evaluation of the terms $\left\langle J_i^a \sigma_i \right\rangle_{dyn}$. In order to do this, one should keep in minds formulas (26)-(29). By definition 93 we have that

$$\langle J^a \sigma \rangle_{dyn} = \int d\sigma \, d\hat{\sigma} \, dJ \, e^{-\frac{1}{2T_a \tau_a} \int d\tau' \left( \tau_a \dot{J}^a + J^a - \gamma_a \sigma \right)^2 + \gamma_a \int d\tau' \sum_a \hat{\sigma}(\tau') J^a(\tau')} J^a \sigma \,, \qquad (99)$$

where for simplicity we omit here the subindex $i$. The integral is Gaussian, we wish again to integrate it by parts. The variation of the exponent is:

$$\frac{1}{T^a \tau_a} \left\{ -(\tau_a)^2 \ddot{J}^a + J^a + \gamma_a \tau_a \dot{\sigma} - \gamma_a \sigma \right\} - \gamma_a \hat{\sigma} =$$
$$\frac{1}{T^a \tau_a} \left\{ \left( -\tau_a \frac{\partial}{\partial \tau} + 1 \right) \left( \tau_a \frac{\partial}{\partial \tau} + 1 \right) J^a - \gamma_a \left( -\tau^a \frac{\partial}{\partial \tau} + 1 \right) \sigma \right\} - \gamma_a \hat{\sigma} \,, \qquad (100)$$

so that we may replace in the average

$$J^a \to \gamma_a \left\{ T^a \tau_a \left( -\tau_a \frac{\partial}{\partial \tau} + 1 \right)^{-1} \left( \tau_a \frac{\partial}{\partial \tau} + 1 \right)^{-1} \hat{\sigma} + \left( -\tau_a \frac{\partial}{\partial \tau} + 1 \right)^{-1} \sigma \right\} \qquad (101)$$

note the sign in the bracket in the last term, which may be adjusted by integrating by parts. Now we use (29) in equation (101) to write the inverses obtaining

$$\langle \sigma J^a \rangle_{dyn} \to \gamma_a \int d\tau' \left\{ \mathcal{C}_a(\tau - \tau') \langle \sigma(\tau) \hat{\sigma}(\tau') \rangle_{dyn} + \mathcal{R}_a(\tau - \tau') \langle \sigma(\tau) \sigma(\tau') \rangle_{dyn} \right\} \,. \qquad (102)$$

Now, reinstating the indices $i$, keeping in mind that $\gamma_a^2 \to (1-t)(q_a - q_{a-1})$ and using (97) we obtain:

$$\left\langle \frac{1}{\sqrt{1-t}} \sum_a \sqrt{q_a - q_{a-1}} \sum_i J_i^a \sigma_i \right\rangle_{dyn} = N \int^\tau d\tau' \left\{ \mathcal{C}(\tau - \tau') R(\tau - \tau') + \mathcal{R}(\tau - \tau') C(\tau - \tau') \right\} \,, \qquad (103)$$

where

$$\mathcal{C} = \sum_a (q_a - q_{a-1}) \mathcal{C}_a \,, \quad \mathcal{R} = \sum_a (q_a - q_{a-1}) \mathcal{R}_a \,. \qquad (104)$$

## 7.3 Dynamical version of Guerra's remainder

Going back to (90) and putting all terms together:

$$-\frac{2}{N} \left\langle \frac{\partial H_t}{\partial t} \right\rangle_{dyn} = -\int^\tau d\tau' \left\{ CR + \mathcal{R}C \right\} (\tau, \tau') + \int^\tau d\tau' \left\{ CR \right\} (\tau, \tau')$$
$$= -\int^\tau d\tau' \left\{ \mathcal{C}\mathcal{R} \right\} (\tau, \tau') + \int^\tau d\tau' \left\{ C - \mathcal{C} \right\} \left\{ R - \mathcal{R} \right\} (\tau, \tau') \,. \qquad (105)$$

This is closely analogous to Guerra's expression for the remainder [14]. To see this bear in mind the connection between statics and dynamics discussed in the previous sections. We consider a process, where the parameters are adiabatically varying. We assume the this process is stationary step by step, and satisfies at each step:

$$
\begin{aligned}
R(\tau) &= -\beta x(\tau) C'(\tau) = -\beta(\tau) C'(\tau), \\
\mathcal{R}(\tau) &= -\beta \tilde{x}(\tau) \mathcal{C}'(\tau) = -\tilde{\beta}(\tau) \mathcal{C}'(\tau),
\end{aligned} \qquad (106)
$$

for two different effective temperatures $\beta(\tau), \tilde{\beta}(\tau)$. One can substitute the above relations in (105). As example one can write

$$\int^\tau d\tau' \, \{\mathcal{C}\,\mathcal{R}\}(\tau,\tau') = -\int_0^\infty d\tau \, \{\mathcal{C}\,\mathcal{C}'\}(\tau)\tilde{\beta}(\tau) = \frac{1}{2}\left(1 + \int_0^\infty d\tau \, \mathcal{C}^2(\tau)\frac{d\tilde{\beta}}{d\tau}\right).$$

When the multithermalization condition $\beta(\tau) = \tilde{\beta}(\tau) = \beta x(\tau)$ holds for all $\tau$ one gets

$$-\frac{1}{N}\left\langle\frac{\partial H_t}{\partial t}\right\rangle_{dyn} = -\frac{\beta}{4}\left(1 + \int_0^\infty d\tau \, \mathcal{C}^2(\tau)\frac{dx}{d\tau}\right) + \frac{\beta}{4}\int_0^\infty d\tau \, \{C - \mathcal{C}\}^2(\tau)\frac{dx}{d\tau}, \quad (107)$$

and since $\frac{dx}{d\tau}$ has *by construction* a negative sign, the negativity of the Guerra's remainder for dynamical average is obtained in dynamical setting with the *assumption* of multithermalization. We may now perform thermodynamic integration of the l.h.s., and because of multithermalization we obtain a dynamical version of the Guerra's bound (86).

The relevance of such positivity in a dynamical setting is still to be understood. On one hand, in the equilibrium picture the positivity of the remainder has provided an excellent guide to search for the rigorous proof of the Parisi solution in the mean field case. On the other hand the dynamical setting described here provides a bridge with experimentally accessible computations and thus makes possible to test the robustness of the positivity property also beyond the assumption of multi-reversible thermalization.

# 8 Being realistic: practical measures for glasses

A realistic glass may be modelled as a system of particles, of different sizes to avoid crystallisation. One may subject such a system to a multibath, by applying uncorrelated fields to each particle, themselves in contact with a slow thermal baths.

On the other hand, several developments in the 90's [10,29,30] based on the Random First Order scenario, pointed to the fact that the structure of an aging glass could be reproduced by the measure (5) with a Hamiltonian

$$H = \sum_{ij} V(\vec{x}_i - \vec{x}_i) + \gamma\sum_i \vec{J}_i \cdot \vec{x}_i + \sum_i |\vec{J}_i|^2, \quad (108)$$

with $\zeta = \frac{T}{T_{eff}}$ and $T_{eff}$ a free parameter, adjusted to represent the out of equilibrium system at its age, of the order of what the temperature was at the moment it fell out of equilibrium.

As one can see, $\zeta$ is not the only parameter because there is also the intensity $\gamma$, and herein lies the entire problem. The auxiliary slow bath selects the states with the appropriate effective temperature, but in order to do so, it needs to have an intensity $\gamma$ that scales with the rate of escape from those states, their inverse lifetime. In a mean-field situation, as explained in Section IV A, in which states with higher free-energy density have an exponentially large lifetime $\sim e^{aN}$, one can let $\gamma \to 0$ at the end of the calculation, because states have zero escape rate in the thermodynamic limit. In other words, the thermodynamic limit and the $\gamma \to 0$ limit do not commute. In a finite-dimensional case, where the escape rate is finite, the limit $\gamma \to 0$ sends us back to the usual Gibbs-Boltzmann measure, and we get nothing. In other words, the thermodynamic limit and the $\gamma \to 0$ limit do not commute. We need a value of $\gamma$ that is as small as possible, but large enough to compensate for the escape rates (i.e. the finite lifetime) of the states. Clearly, the construction is not without ambiguities.

When we work with a multibath, we make the same construction, and of course we have the same problem with $\gamma$. However, here we have a direct experimental test of our assumptions. Consider the following protocol: we let our system age. Assume that at time $t_w$ it is

still out of equilibrium. Regardless of whether it will eventually equilibrate or not, we wish to characterize the measure that describes it such as it is. We check by measuring correlation and response that the system has a two-temperature fluctuation-dissipation behavior, a fact that is well-attested numerically, at least as a good approximation. Now we apply a weak multibath such that: *i)* timescale is of the order of the $\alpha$ correlation decay (from $q_{EA}$ to zero) scale of the glass, *ii)* it has the same temperature as the fluctuation-dissipation one of the system, and crucially, *iii)* its amplitude is just sufficient so that we verify that the system becomes stationary by virtue of its interaction with the bath (the $\alpha$ timescale ceases to grow, as it does in an aging system). Next, we slowly change parameters of multibath, temperature and intensity $\gamma$, always verifying that the timescale of the slow bath is of the order of the $\alpha$ timescale, and that the effective slow temperatures of bath and system are the same. If such a procedure is possible, and we can take the system to the liquid situation in which it is in ordinary equilibrium, then a (multi)thermodynamic integration is legitimate, and we have in effect experimentally proven, following the results in the previous sections, that the system as it was at the 'age' at which we started, may indeed be described by the multibath measure, with the amplitude we needed to ascribe to it so that the system remained stationary from the moment we connected the multibath, and was multithermalized by it.

Note that it is both the parameters $T_{eff}$ and $\gamma$ that play a role. Indeed, we may describe this as a (Gedanken) experiment to measure these two parameters.

## 9 Conclusions

The Parisi construction, although often described as *a solution*, is in fact something wider: a symmetry-breaking scheme [32]. In this sense it is more akin to the general ideas of ferromagnetism from Curie to Landau, than to the Onsager solution for the Ising model. This is why we may ask if it applies to other problems, such as finite-dimensional spin glasses. Indeed, the fact that it is a scheme where a symmetry group is broken into symmetry subgroups means that it is possible to propose a solution of this form *in any model*. The problem is, however, that the symmetry in question (replica symmetry) is a very bizarre one, and is too closely dependent on one particular formalism.

On the other hand, the dynamics with many timescales – and one temperature scale at each – can be thought of as as a symmetry-breaking situation as well. Consider first ordinary thermalization of Hamiltonian dynamics. When a system is thermalized at a given temperature and then isolated, it has a dynamic time-reversal symmetry with temperature as its parameter, that implies the fluctuation-dissipation and Onsager reciprocity relations. The Hamiltonian dynamics itself, before choosing a temperature, had a larger group of symmetry, and thermalization can be seen as the act of breaking down this symmetry to a subgroup labeled by the temperature. Multithermalization of widely separated timescales corresponds to a more complicated breaking of the large symmetry group, with a parameter for each timescale. This may seem an unnecessarily pompous and abstract way of putting things, but, again, it allows us to see that any system with slow dynamics and widely separated timescales may possess a solution with multithermalization.

## Acknowledgments

J.K. is supported by the Simons Foundation Grant No 454943, E.M. is partially supported by Almaidea Grant 2017.

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
