# Peer review of "Stationarization and Multithermalization in spin glasses"

_SciPost Physics, doi:SciPost Phys. 10, 113 (2021)_

## Round 1 · Referee Report · Anonymous (Referee 2) · 2021-2-25

Strengths

The subject is potentially of high interest.

Weaknesses

The paper does have a proper introduction, and very short conclusion. It is very difficult to get any message from the paper.

Report

This paper concerns slow dynamics in spin glasses and develops an old idea that the Replica number n, as well as the Parisi breaking parameters, called $\zeta_i$ in the paper, can be associated to (ratios of) effective temperatures of degrees of freedom evolving over well separated time scales. The authors include in their system slow degrees of freedom, evolving on hierarchically organised time scales and coupled with baths at different temperatures. This is what they call a multibath.

The thesis of the paper, as stated in the abstract, is that systems like spin glasses that fail to equilibrate when in contact with a simple bath may equilibrate when in contact with a properly chosen multibath. This is of course only possible when one can neglect heat transfer between degrees of freedom that evolve on different scales that hence should be widely separated, and should be distinguished from generic stationarisation leading to off-equilibrium states. Thermalization occurs if, in addition to stationarization, the temperatures of the multibath coincide with the ‘effective temperatures’ in fluctuation-dissipation relations of all possible quantities.

The authors consider the soft and hard spin SK model at low temperature as an example of a system that could be multithermalized, if the multibath temperatures are tuned to the effective temperatures that spontaneously rise in the system. Multithermalization would also be possible in any hypothetical finite dimensional systems with replica symmetry breaking.

They suggest that the multibath approach can be used to define a dynamic analogous of the interpolating Hamiltonian used by Guerra in his RSB bounds. Simulations of the SK model in contact with a multibath are presented.

Despite a high potential interest, I found the paper is quite confused. It hard to grasp a clear line of thought. The reading of the paper is very difficult, not (only) for the concept involved, but for the rather chaotic way they are exposed. In fact, after many reading of the paper I fail to understand what is the message that the authors wish to convey. I think that any interested readers would be grateful if the paper is rewritten in a more linear way. The lack of a proper introduction, as well as a clear concluding section does not help either to find a motivation to proceed with the reading of the paper or to orient in the various arguments of the different section.

I suggest a reorganisation of the paper, with a clear introduction, where the problem addressed in the paper is stated, the theoretical approach described, and the main result discussed. In the same way, the paper should contain a proper concluding section with a summary of the results and the perspectives opened by the paper.

Moreover, the simulations the way they are presented in this paper do not meet the minimal standards to be published in a physics journal. In the present form it almost lacks any quantitative analysis on the presented data, and it is impossible to imagine how the figures are representative of generic/general situations. A much more systematic study is needed to be convincing of anything.

Said that, the paper rises many questions: in a generic system in contact with several baths at different temperatures, heat would flow from warm to cold baths. Dynamically, multithermalization can only occur in a limiting sense if the time scales of the baths have an extreme separation and heat flows tend to zero. The authors of course understand this, and mention time separation several times in the paper. Unfortunately however in several places, they give the impression that multithermalization can occur in finite times. For example in the abstract they write “We consider those systems that do not thermalize in finite times when in contact with an ordinary bath but may do so in contact with a multibath.” and in section IV: “It may furthermore multithermalize in times of order one, but if and only if T1 = T threshold .” and “the timescale-separations required for the multibath, though large, need not diverge in the thermodynamic limit, if the system is finite-dimensional.” etc.

One needs to start with finite time scales to achieve stationarization, and in that case heat would flow. The limit of infinite time scale should be taken next to reduce heat flows to zero. This is very much like in (mean-field) ageing except that there is no multi-bath and the effective temperatures, as well as the infinite time scales emerge spontaneously. Similarities between slow stationarization and ageing were previously investigated by one of the authors. It is not clear what is the novelty here -except maybe to pass from 1RSB to full-RSB-.

A related point is the characterisation of the system that can multithermalize. Is it true that the systems that can multithermalize are the ones like the SK model, that develop their own effective temperatures? What is the role of time-scales and emerging time-reparametrization invariance? And the strength of the coupling between fast and slow variables? In sec. II C it would seem that the multibath could ‘impose’ its temperature to the system. How general is this? What is the interest in ‘imposing’ an external temperature?

Concerning the Guerra interpolation: in equilibrium the Guerra method has played a fundamental role in the mathematical comprehension of mean-field spin glasses. In dynamics, proofs of the dynamical mean field equations exist, which do not use the to interpolation method. One would expect however that the method could be adapted to provide alternative proves of the dynamical mean-filed equations. This is not the use that the authors do, as they are interested to the multi-thermalized situation, that as observed is only achivable in a limiting sense. In fact they focus on
the Guerra remainder and show that this is positive under the hypothesis of multithermalization.
The questions then are (1) multithermalization being only possible in a limiting sense the hypothesis is highly non-trivial, not just an innocent hypotheis as suggested in the text. (2) why
to care about the interpolating model in dynamics? (3) why should we be happy if the remainder is positive? Is it only positive for large times? etc.

Summarising, I think that the paper might give rise to interesting developments, but that a complete revision of principle, clarifying the main concepts which concepts should be conveyed to the reader, technical: presenting the simulations in a systematic and quantitative way, and of form is necessary before publication.

Requested changes

Completely rewrite in a linear and clear way.

  • validity: -
  • significance: good
  • originality: high
  • clarity: poor
  • formatting: -
  • grammar: acceptable

Author:  Emanuele Mingione  on 2021-04-06  [id 1344]

(in reply to Report 2 on 2021-02-25)

We thank the Referee for the careful report provided and for having highlighted points which deserved more attention. We have followed the recommendations and we believe that the paper has substantially improved in clarity. A list of the modified points is in the attached file

The Authors

Attachment:

ListModifications.pdf

---

## Round 1 · Referee Report · Anonymous (Referee 1) · 2021-2-25

Strengths

1- general strategic idea 2- implementation for mean field spin glasses 3- application to broken replica symmetry bound 4- new interpretation of replica symmetry breaking 5- dynamical setting 6- hints for further developments

Weaknesses

1- The multiplicity and variety of applications is only an apparent weakness. As a matter of fact the whole field is stabilising as a completely new and innovative general frame to treat disordered statistical mechanics systems on sound dynamical basis.

Report

referee report on the paper

Stationarization and Multithermalization in spin glasses
by
Pierluigi Contucci, Federico Corberi, Jorge Kurchan, and Emanuele Mingione

In this paper the Authors continue their investigation on the behaviour of statistical mechanics systems in contact with a multibath having different temperatures and widely separated timescales. These multibaths are strongly connected with the nested Gibbs-Boltzmann structures exploited in the treatment of mean field disordered systems in the frame of the replica trick, or the broken replica symmetry bounds.
The peculiar treatment adopted by the Authors allows to consider these structures in a dynamical framework. In particular it is shown how a Langevin dynamics for external magnetic fields, interacting with the system, produces effects equivalent to a multibath. Time correlations and response functions for the multibath are also studied. Applications to the Sherrington-Kirkpatrick model for mean field spin glasses are extensively developed, and the relations with the Parisi partial differential equation and the functional order parameter are established. In the general dynamical frame for the SK model, the Authors consider an extremely interesting dynamical frame where both the \sigma spin variables and the J variables evolve in time according to well definite laws, such that at the equilibrium the system reaches a distribution connected with the \zeta number of replicas.
This general scheme allows also the study of stationarization and thermalisation in various cases, even in the thermodynamic limit. The T-\zeta phase diagram arising in the replica trick
(T temperature, \zeta number of replicas) is given a general physical picture in terms of a two temperature multibath.
The interpolation scheme of the broken replica bounds is given a dynamical interpretation, with a new expression of the error term.
The very definition of replica symmetry breaking is given a new dynamical interpretation.

In conclusion, the analysis developed in this paper gives further evidence that ideas based on the multibath can be very useful in the study of disordered systems. This is a real seminal paper, potentially suggesting new useful developments.
I strongly suggest publication of this paper on SciPost.

Requested changes

No specific changes are requested.

---

## Round 2 · Referee Report · Anonymous · 2021-4-28

Report

The resubmitted papers presents marginal modifications address the points I raised. A short and not very informative introduction (comprising 9 lines and a summary of the content of the sections), and a few sentences have been added. The only criticism they addressed in a satisfactory way was to give a professional description of their simulations (though probably the authors did not simulate systems of 10^12 spins as they claim). In deciding for this minimalistic strategy rather than going through a deep revision as I suggested the authors have probably lost an occasion to make their work appreciable by a wider community. I do not oppose myself to the publications.

---

## Editorial Decision

published